# Block-Biased Mamba for Long-Range Sequence Processing

**Annan Yu**
Center for Applied Mathematics
Cornell University
Ithaca, NY 14853
ay262@cornell.edu

**N. Benjamin Erichson**
Lawrence Berkeley National Laboratory
International Computer Science Institute
Berkeley, CA 94720
erichson@icsi.berkeley.edu

## Abstract

Mamba extends earlier state space models (SSMs) by introducing input-dependent dynamics, and has demonstrated strong empirical performance across a range of domains, including language modeling, computer vision, and foundation models. However, a surprising weakness remains: despite being built on architectures designed for long-range dependencies, Mamba performs poorly on long-range sequential tasks. Understanding and addressing this gap is important for improving Mamba's universality and versatility. In this work, we analyze Mamba's limitations through three perspectives: expressiveness, inductive bias, and training stability. Our theoretical results show how Mamba falls short in each of these aspects compared to earlier SSMs such as S4D. To address these issues, we propose $B_2S_6$, a simple extension of Mamba's S6 unit that combines block-wise selective dynamics with a channel-specific bias. We prove that these changes equip the model with a better-suited inductive bias and improve its expressiveness and stability. Empirically, $B_2S_6$ outperforms S4 and S4D on Long-Range Arena (LRA) tasks while maintaining Mamba's performance on language modeling benchmarks.

## 1 Introduction

Mamba [25] has recently emerged as an exciting alternative to Transformers, demonstrating strong empirical performance not only in language tasks [84, 45], but also in a variety of domains, including vision [107, 103], audio [43], time series forecasting [43], and operator learning [33]. Based on the state-space model (SSM) framework of S4 and S4D, Mamba replaces attention with a selective state space mechanism. Unlike attention, which computes pairwise interactions across the sequence, this approach uses data-dependent weights in a recurrent structure, aligning with the temporal nature of sequential data. Despite its success as a general-purpose sequence modeling tool [51], Mamba consistently underperforms on benchmarks such as the Long-Range Arena (LRA) [3]. This limitation is surprising, given that Mamba is derived from architectures specifically designed for modeling long-range dependencies. This behavior raises an important question:

*Why does Mamba struggle with long-range sequence modeling?*

To address this question, we conduct a theoretical analysis of the S6 state-space unit that underlies the Mamba architecture. Our goal is twofold: first, to explain why Mamba underperforms on long-range sequence tasks, and second, to provide a principled remedy. Through a rigorous mathematical analysis, we identify three key factors that limit Mamba's ability to model long-range dependencies:

- **Expressiveness.** Unlike S4D, which allows each internal channel to learn an independent recurrent unit, Mamba shares parameters across all channels. Therefore, one should not interpret Mamba as

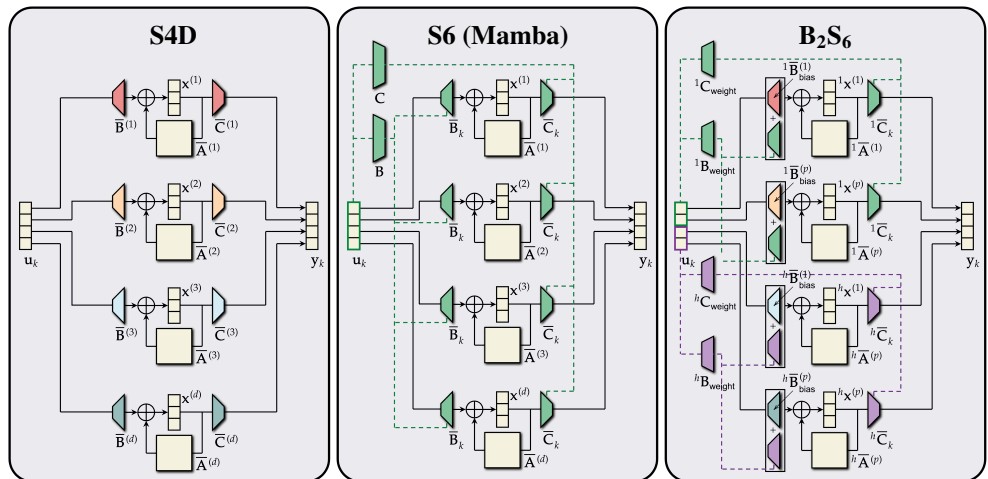

Figure 1: Comparison of S4D, S6, and $B_2S_6$ units. S4D uses independent linear SSM units for each channel, giving it high capacity (or width) but no input-dependent selectivity. S6 introduces a selective mechanism that modulates its internal dynamics based on the input, but shares parameters across channels, limiting its effective width and expressiveness. Our proposed $B_2S_6$ unit partitions the input into smaller blocks, enabling selective behavior across multiple subspaces (see also [20]). Additionally, it includes a channel-specific, input-independent bias term that further increases model capacity. These design choices improve the performance on long-range sequence tasks.

a straightforward extension of S4D that gains time-variant benefits at no cost. Rather, Theorem 1 shows that while Mamba introduces input-dependent dynamics, it also sacrifices certain degrees of freedom. Most notably, it has a weaker ability to learn independent behavior in each channel.

- **Inductive Bias.** Unlike S4D, which uses a position-based bias that helps preserve long-range memory, Mamba adaptively decides what information to remember or forget based on the input. This input-dependent mechanism is too extreme and can cause the model to discard useful long-term information too quickly, leading to poor retention in long-range tasks (see Theorem 2).

- **Training Stability.** Unlike S4D, which tends to train reliably on long sequences, Mamba empirically exhibits unstable training behavior on the LRA benchmark tasks. We theoretically show that this instability comes from the input-dependent selection mechanism, which becomes increasingly difficult to optimize as the length of the sequence increases (see Theorem 3).

Motivated by these findings, we propose the *Block-Biased-S6* unit ($B_2S_6$), a simple yet principled extension of the S6 architecture that addresses its limitations in modeling long-range dependencies (see Figure 1). Like S6, $B_2S_6$ retains the selective mechanism that modulates internal dynamics based on the input. However, instead of computing the selection weights from the full input $\mathbf{u}_k$, we divide the model into smaller blocks, with each block operating on a subset of $\mathbf{u}_k$. This multihead structure was already suggested in [20], but in addition to that, we introduce an input-independent, channel-specific bias term to the recurrent unit (see Table 1). We prove that these modifications enhance $B_2S_6$'s expressiveness and assign it a more appropriate inductive bias for handling long-range dependencies. We also apply a reduced learning rate to the parameters governing the input-dependent sampling intervals, which improves training stability. Together, these modifications enable $B_2S_6$ to model long-range dependencies more effectively. On the LRA benchmark, $B_2S_6$ resurrects Mamba from failure and even outperforms S4 and S4D. Furthermore, we show that $B_2S_6$ achieves comparable perplexity to Mamba when trained on language tasks, demonstrating its versatility across domains.

**Contributions.** Our main contributions are as follows:

1. We analyze the limitations of Mamba in modeling long-range dependencies and provide theoretical results that characterize the failure modes from three key aspects: (i) expressiveness (see Theorem 1), (ii) inductive bias (see Theorem 2), and (iii) training stability (see Theorem 3).

2. We introduce the $B_2S_6$ unit, a principled extension of the S6 architecture. We show that $B_2S_6$ has universal approximation capabilities (see Theorem 4) and is thus more expressive, while also equipped with a more suitable inductive bias for long-range sequence processing (see Theorem 5).

Table 1: A comparison of different classes of selective or non-selective state space models.

| | S4D | S5 | Mamba | Mamba2 | B2S6 |
|---|---|---|---|---|---|
| $\Delta_t$ Dependency | input-independent | input-independent | input-dependent | input-dependent | input-dependent |
| B/C Dependency | input-independent | input-independent | input-dependent | input-dependent | both |
| Number of Heads | $\geq 1$ | $\geq 1$ | $= 1$ | $\geq 1$ | $\geq 1$ |
| Parameters | complex | complex | real | real | complex |
| Parameterization of A | diagonal | diagonal | diagonal | scalar | diagonal |

3. We evaluate $B_2S_6$ on the Long-Range Arena benchmark, where it achieves state-of-the-art performance. In addition, a preliminary examination on the SlimPajama dataset [78] shows that $B_2S_6$ matches Mamba's performance on language modeling tasks, demonstrating its versatility.

## 2 The S4D and S6 recurrent units

Introduced in [28] and [27], the S4 and S4D recurrent units process sequences of $d$-dimensional inputs $\mathbf{u} = (\mathbf{u}_1, \ldots, \mathbf{u}_L)$ to produce corresponding $d$-dimensional outputs $\Gamma_{\text{S4/S4D}}(\mathbf{u}) = \mathbf{y} = (\mathbf{y}_1, \ldots, \mathbf{y}_L)$. Throughout this paper, we use subscripts to index positions in a sequence and superscripts to index channels. For example, $\mathbf{u}_k^{(i)}$ denotes the $i$th entry of the $k$th input vector $\mathbf{u}_k$. Each channel of the S4/S4D model is computed independently with a linear system. For the $i$th channel, the update rule is

$$\mathbf{x}_k^{(i)} = \overline{\mathbf{A}}^{(i)} \mathbf{x}_{k-1}^{(i)} + \overline{\mathbf{B}}^{(i)} \mathbf{u}_k^{(i)}, \quad \mathbf{y}_k^{(i)} = \overline{\mathbf{C}}^{(i)} \mathbf{x}_k^{(i)}, \qquad 1 \leq i \leq d, \tag{1}$$

where $\mathbf{x}_k^{(i)} \in \mathbb{C}^{n \times 1}$ is the hidden state with initial condition $\mathbf{x}_0^{(i)} = \mathbf{0}$, and $\mathbf{u}_k^{(i)} \in \mathbb{R}$. The matrices $\overline{\mathbf{A}}^{(i)} \in \mathbb{C}^{n \times n}$, $\overline{\mathbf{B}}^{(i)} \in \mathbb{C}^{n \times 1}$, and $\overline{\mathbf{C}}^{(i)} = \mathbb{C}^{1 \times n}$ are derived from a continuous-time system. While there are many different discretization rules, we focus on the Zero-Order Hold (ZOH) discretization:

$$\overline{\mathbf{A}}^{(i)} = \exp(\Delta^{(i)} \mathbf{A}), \quad \Delta^{(i)} = \exp(b^{(i)}), \qquad \overline{\mathbf{B}}^{(i)} = \mathbf{A}^{-1}(\overline{\mathbf{A}}^{(i)} - \mathbf{I})\mathbf{B}^{(i)}, \quad \overline{\mathbf{C}}^{(i)} = \mathbf{C}^{(i)}. \tag{2}$$

Here, $\mathbf{A} \in \mathbb{C}^{n \times n}$, $\mathbf{B}^{(i)} \in \mathbb{C}^{n \times 1}$, $\mathbf{C}^{(i)} \in \mathbb{C}^{1 \times n}$, and $b^{(i)} \in \mathbb{R}$ are trainable parameters, specific to each channel $1 \leq i \leq d$.[1] S4 and S4D are particularly well-suited for modeling sequences with long-range dependencies and achieve state-of-the-art results on the Long-Range Arena benchmark. This strength is largely due to the reparameterized $\Delta^{(i)}$ values, which, when chosen small, give the model long-term memory by slowing the system dynamics. S4 and S4D differ only in how the matrix $\mathbf{A}$ is represented; we henceforth focus on S4D, which uses a diagonal $\mathbf{A}$ consistent with Mamba.

One limitation of the S4D units is that their dynamics are linear and time-invariant. The S6 units [25] used in the Mamba models improve upon this by making the dynamics input-dependent. More specifically, given a $d$-dimensional input sequence $\mathbf{u} = (\mathbf{u}_1, \ldots, \mathbf{u}_L)$, an S6 unit computes the $i$th channel of the output $\Gamma_{\text{S6}}(\mathbf{u}) = \mathbf{y} = (\mathbf{y}_1, \ldots, \mathbf{y}_L)$ as follows:

$$\mathbf{x}_k^{(i)} = \overline{\mathbf{A}}_k^{(i)} \mathbf{x}_{k-1}^{(i)} + \overline{\mathbf{B}}_k^{(i)} \mathbf{u}_k^{(i)}, \quad \mathbf{y}_k^{(i)} = \overline{\mathbf{C}}_k \mathbf{x}_k^{(i)}, \qquad 1 \leq i \leq d, \tag{3}$$

where $\mathbf{x}_k^{(i)} \in \mathbb{R}^n$ and $\mathbf{u}_k^{(i)}, \mathbf{y}_k^{(i)} \in \mathbb{R}$. The matrices $\overline{\mathbf{A}}_k^{(i)}$, $\overline{\mathbf{B}}_k^{(i)}$, and $\overline{\mathbf{C}}_k$ are computed at each step as

$$\overline{\mathbf{A}}_k^{(i)} = \exp(\Delta_k^{(i)} \mathbf{A}), \quad \Delta_k^{(i)} = \text{softplus}(\mathbf{w}^\top \mathbf{u}_k + b^{(i)}), \quad \overline{\mathbf{B}}_k^{(i)} = \mathbf{A}^{-1}(\overline{\mathbf{A}}_k^{(i)} - \mathbf{I})\mathbf{B}\mathbf{u}_k, \quad \overline{\mathbf{C}}_k = \mathbf{u}_k^\top \mathbf{C}. \tag{4}$$

The trainable parameters of the model are $\mathbf{A} \in \mathbb{R}^{n \times n}$, $\mathbf{B} \in \mathbb{R}^{n \times d}$, $\mathbf{C} \in \mathbb{R}^{d \times n}$, $\mathbf{w} \in \mathbb{R}^d$, and $b^{(i)} \in \mathbb{R}$ for all $1 \leq i \leq d$.[2] The key innovation in the S6 architecture lies in the input-dependent dynamics. Both $\overline{\mathbf{B}}_k$ and $\overline{\mathbf{C}}_k$ change with each input step, introducing nonlinearity into the system. In addition, the sampling interval $\Delta^{(i)}$ is also input-dependent, allowing the model to vary the rate at which it memorizes or forgets information. Comparing eq. (4) to eq. (2), we note that $\mathbf{B}\mathbf{u}_k$ and $\mathbf{u}_k^\top \mathbf{C}$ in an S6 are shared across all channels, while $\mathbf{B}^{(i)}$ and $\mathbf{C}^{(i)}$ in an S4D are channel-specific. In the next section, we show how this distinction limits the "effective width" and expressiveness of an S6 unit.

---

[1]Note that we tie the matrix $\mathbf{A}$ across all channels, which is consistent with the LRA training setup. This choice can be loosened without affecting the essence of the paper.

[2]In many implementations of Mamba, the selection of $\Delta_k^{(i)}$ is more complex, and $\overline{\mathbf{B}}_k$ may be computed using a forward Euler method instead. We follow the formulation in Algorithm 2 of the original Mamba paper [25].

# 3 A single-layer Mamba is not a universal approximator

A universal approximation theorem (UAT) characterizes the expressive power of neural networks by showing that, under certain conditions, they can approximate any target function to arbitrary accuracy, provided they are sufficiently wide or deep [60, 39]. While UATs do not directly address training dynamics, they serve as a useful sanity check for a model's theoretical capacity. For deep S4D models, some universal approximation results have been established in [87]. In this section, we prove a new UAT of S4D and a new non-UAT of S6 that highlight how the shared state matrices in S6 significantly constrain its expressiveness. We investigate the task of learning a univariate sequential task defined by a ground-truth function $G$, using a single-layer model that takes in an input $(u_1, \ldots, u_L)$, linearly embeds the input into a $d$-dimensional space, passes it through an S4D or S6 unit, and then decodes the output. More precisely, we consider the class of models (see Figure 5) defined by

$$\tilde{G}(\mathbf{u}) = \tilde{G}((u_1, \ldots, u_L)) = \mathbf{N}\sigma(\Gamma((\mathbf{M}u_1, \ldots, \mathbf{M}u_L))_L + \boldsymbol{\theta}), \tag{5}$$

where $\mathbf{M} \in \mathbb{R}^{d \times 1}$ is an encoder, $\mathbf{N} \in \mathbb{R}^{d \times 1}$ is a decoder, $\boldsymbol{\theta} \in \mathbb{R}^{d \times 1}$ is a bias term, and $\sigma : \mathbb{R} \to \mathbb{R}$ is a nonlinear activation function applied entrywise to the last output of an S4D or S6 unit $\Gamma$. There is only one remaining issue: in the definition of S6, we have $\overline{\mathbf{C}}_k = u_k^\top \mathbf{C}$. That means $\Gamma_{\text{S6}}((\mathbf{M}u_1, \ldots, \mathbf{M}u_L))_L$ is zero as long as $u_L = 0$, which makes $\tilde{G}$ with $\Gamma_{\text{S6}}$ clearly not universal approximators on the domain $[0, 1]^L$. To avoid "cheating" in this way, we assume that $u_L = 1$ and only focus on approximating functions on $[0, 1]^{L-1} \times \{1\}$, which makes our non-UAT stronger.

**Theorem 1.** The single-layer S4D models are universal approximators of continuous functions, but the single-layer S6 models are not. More precisely, fixing a constant for $\Delta^{(i)}$ for all $1 \leq i \leq d$ in eq. (2) and $\Delta_k^{(i)}$ for all $1 \leq i \leq d$ and $1 \leq k \leq L$ in eq. (4), the following two statements hold:

1. Let $\sigma : \mathbb{R} \to \mathbb{R}$ be any Lipschitz continuous, non-polynomial activation function. Given any continuous function $G : [0, 1]^L \to \mathbb{R}$ and any $\epsilon > 0$. There exist some $d \geq 1$, $n \geq 1$, and a choice of parameters $\mathbf{M}, \mathbf{N}, \boldsymbol{\theta}, \mathbf{A}, \mathbf{B}^{(i)}$, and $\mathbf{C}^{(i)}$, where $1 \leq i \leq d$, such that the map $\tilde{G}$ in eq. (5) with $\Gamma = \Gamma_{\text{S4D}}$ in eq. (1) satisfies that $|\tilde{G}(\mathbf{u}) - G(\mathbf{u})| \leq \epsilon$ for any $\mathbf{u} \in [0, 1]^L$.

2. Let $\sigma : \mathbb{R} \to \mathbb{R}$ be any function and let $L \geq 3$ be given. There exists a continuous function $G : [0, 1]^{L-1} \times \{1\} \to \mathbb{R}$ and some $\epsilon > 0$ such that for any $d \geq 1$, any $n \geq 1$, and any choice of parameters $\mathbf{M}, \mathbf{N}, \boldsymbol{\theta}, \mathbf{A}, \mathbf{B}$, and $\mathbf{C}$, the map $\tilde{G}$ in eq. (5) with $\Gamma = \Gamma_{\text{S6}}$ in eq. (3) satisfies that $|\tilde{G}(\mathbf{u}) - G(\mathbf{u})| > \epsilon$ for some $\mathbf{u} \in [0, 1]^{L-1} \times \{1\}$.

You should not interpret Theorem 1 as a pessimistic claim that Mamba models are incapable of solving complex tasks. In practice, Mamba architectures typically use multiple stacked layers and do not fix $\Delta_k^{(i)}$, both of which enhance the expressiveness. Rather, it highlights that an S6 unit is not a simple extension of S4D that inherits all of its structural principles, and the sharing of the $\mathbf{B}$ and $\mathbf{C}$ matrices across all channels can reduce its capacity, making it potentially less effective than S4D for certain tasks. Beyond its impact on approximation capacity, this also leads to issues in larger models, such as training instability [23, 16, 67], reduced generalization [5, 92, 11, 100, 40], and challenges in interpretability [104, 94].

While Theorem 1 assumes that $\Delta$ is fixed, we use a synthetic experiment to demonstrate that even when $\Delta$ is trainable, the channel-independent design of $\mathbf{B}$ and $\mathbf{C}$ in S6 limits its ability to solve certain problems than S4D. Consider the input function $u(t; \boldsymbol{g}) = \sum_{i=1}^{10} g_i \cos(p_i t)$, where $p_1, \ldots, p_{10}$ are the 10 smallest prime numbers and $\boldsymbol{g} \in \mathbb{R}^{10}$ is a coefficient vector. The learning task is to recover $\boldsymbol{g}$ from the function $u(t; \boldsymbol{g})$, i.e., to learn the mapping $G : u(t; \boldsymbol{g}) \mapsto \boldsymbol{g}$. Intuitively, if the model is sufficiently wide, each channel can specialize in capturing one of the cosine components, making the problem easy. We train both a single-layer S6 model and a single-layer S4D model to approximate this mapping, also allowing the $\Delta$-related parameters to be learned in both cases. As shown in Figure 2, the performance of the S6 model remains nearly constant as the width $d$ increases, reflecting the fact that its effective width does not scale with $d$. In contrast, the S4D model benefits significantly from increased width.

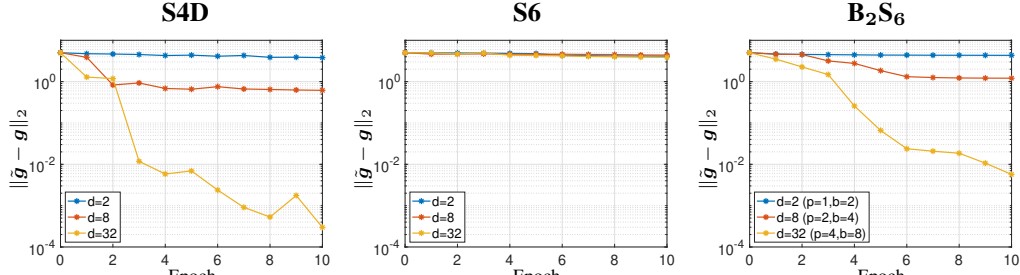

Figure 2: The mean loss $\|\tilde{g} - g\|_2$ between the true coefficient $g$ and the model prediction $\tilde{g}$. Every model has a single layer and is trained for 10 epochs. Here, $d$ is the number of channels in a model. For the $B_2S_6$ model, $h$ is the number of blocks and $p$ is the number of channels in each block.

## 4 Mambas exhibit strong inductive bias

A necessary condition for a model to obtain the capability of handling long-range dependency is that it possesses long-term memory. In S4D models, this is achieved by learning small values of $\Delta^{(i)}$, which slow down the evolution of the continuous-time LTI systems and allow information to persist over longer time horizons [74]. In contrast, S6 models introduce an input-dependent sampling interval $\Delta_k^{(i)}$, where certain inputs lead to larger values of $\Delta_k^{(i)}$, causing faster memory decay, while others result in smaller values and slower decay.

This input-adaptive memory control serves as a useful inductive bias in language modeling, where only a limited number of words in a sentence typically carry semantic significance. However, this same mechanism is less suitable for many LRA tasks involving non-linguistic sequences, such as Image, Pathfinder, and PathX, where inputs are derived from flattened pixel arrays. In these settings, while some positional bias may exist (e.g., central pixels may be more informative than peripheral ones), it is difficult to justify that pixels of certain colors in a CIFAR image should receive dramatically more attention than others. As a result, the input-dependent memory dynamics of S6 may act as an overly aggressive or misaligned inductive bias in such contexts.

To formalize the notion of input-dependent bias, let $\Gamma$ denote either an S4D or S6 unit as defined in eq. (1) and eq. (3), respectively. Given a $d$-dimensional input sequence $\mathbf{u} = (\mathbf{u}_1, \ldots, \mathbf{u}_L)$, we ask: how does the output of $\Gamma$ depend on each individual input vector $\mathbf{u}_k$? To study this, we examine the relative gradient of the output with respect to each $\mathbf{u}_k$. Specifically, we define:

$$S_k = \frac{\|\mathbf{J}_k\|_F}{\sum_{k'=1}^{L} \|\mathbf{J}_{k'}\|_F}, \qquad \mathbf{J}_{k'} = \frac{\partial}{\partial \mathbf{u}_{k'}} \Gamma(\mathbf{u}_1, \ldots, \mathbf{u}_L)_L \in \mathbb{R}^{d \times d}, \qquad (6)$$

where $\|\cdot\|_F$ denotes the Frobenius norm, and $\mathbf{J}_{k'}$ is the Jacobian of the final output vector with respect to the $k'$-th input vector. Intuitively, the quantity $S_k$ measures the relative influence of $\mathbf{u}_k$ on the final output. In other words, it tells us how much of the output's sensitivity to perturbations is attributed to the $k$-th input. For a linear system such as S4D, the Jacobians $\mathbf{J}_{k'}$ are independent of the input $\mathbf{u}$, so the relative gradients $S_k$ remain constant across different inputs. This indicates that the model treats all inputs uniformly and does not introduce bias toward a particular input *as $\mathbf{u}$ changes*. In contrast, we now show that S6 units introduce a bias when input vectors vary in magnitude.

**Theorem 2.** An S6 unit imposes an exponentially large bias as the input magnitude increases. That is, fix a $k_0 \leq L$ and let $\mathbf{A} \in \mathbb{R}^{n \times n}$ be a diagonal matrix with negative diagonal entries and $\mathbf{w} \neq \mathbf{0}$. For almost every (a.e.) input sequence $\mathbf{u} = (\mathbf{u}_1, \ldots, \mathbf{u}_L) \in \mathbb{R}^L$ and a.e. parameters $\mathbf{B} \in \mathbb{R}^{n \times d}$, $\mathbf{C} \in \mathbb{R}^{d \times n}$, and $\mathbf{b} \in \mathbb{R}^d$, let $S_{S6,k}$ be defined in eq. (6), where $\Gamma = \Gamma_{S6}$ is defined in eq. (3). As $c \to \infty$, the following statements hold for any $p > 0$:

1. If $\mathbf{w}^\top \mathbf{u}_{k_0} > 0$, we have

$$S_{S6, k}((\mathbf{u}_1, \ldots, \mathbf{u}_{k_0-1}, c\mathbf{u}_{k_0}, \mathbf{u}_{k_0+1}, \ldots, \mathbf{u}_L)) = \begin{cases} \mathcal{O}(c^{-p}) & , \ k < k_0, \\ \Theta(c^{-1}) & , \ k = k_0, \\ \Theta(1) & , \ k > k_0. \end{cases}$$

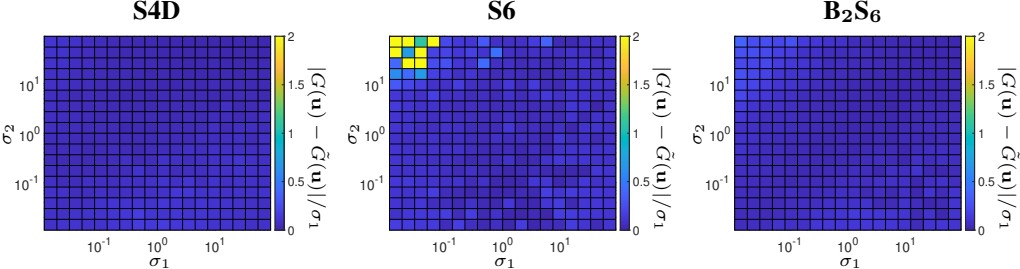

Figure 3: The mean relative loss $|G(\mathbf{u}) - \tilde{G}(\mathbf{u})|/\sigma_1$ for different choices of $\sigma_1$ and $\sigma_2$. The S6 model cannot make useful predictions when $\sigma_1$ is small and $\sigma_2$ is large; $B_2S_6$ fixes this. In all experiments, we fix $d = 32$. For $B_2S_6$, we set $h = 8$ and $p = 4$.

2. If $\mathbf{w}^\top \mathbf{u}_{k_0} < 0$, we have

$$S_{\text{S6},\,k}((\mathbf{u}_1, \ldots, \mathbf{u}_{k_0-1}, c\mathbf{u}_{k_0}, \mathbf{u}_{k_0+1}, \ldots, \mathbf{u}_L)) = \begin{cases} \Theta(1) & , \ k \neq k_0, \\ \mathcal{O}(c^{-p}) & , \ k = k_0. \end{cases}$$

Moreover, let $S_{\text{S4D},k}$ be defined in eq. (6), where $\Gamma = \Gamma_{\text{S4D}}$ is defined in eq. (1). We have that $S_{\text{S4D},k}$ is constant and does not depend on the input $\mathbf{u}$.

The theorem shows that as the magnitude of a single input vector $\mathbf{u}_{k_0}$ grows, one of two extreme behaviors can emerge: either the current input (Case 2) or all previous inputs (Case 1) exert an exponentially diminishing influence on the final output. Neither outcome is desirable when modeling sequences with long-range dependencies. In practice, many implementations of Mamba mitigate this issue through normalization layers that constrain the magnitudes of input vectors. Additionally, unlike the original formulation in [25], more sophisticated parameterizations are often used to compute the sampling intervals $\Delta_k^{(i)}$. These modifications can be interpreted as efforts to counteract the exponential input bias introduced by large-magnitude vectors.

From Theorem 2, we see another perspective of the limitation of S6: in the computation of $\Delta_k^{(i)} = \text{softplus}(\mathbf{w}^\top \mathbf{u}_k + b^{(i)})$, the vector $\mathbf{w}$ defines two half-spaces in $\mathbb{R}^d$. Inputs $\mathbf{u}_k$ that lie in the positive half-space produce larger values of $\Delta$, leading to slower dynamics and better memorization, whereas those in the negative half-space yield smaller $\Delta$, resulting in faster forgetting. This setup restricts the model's ability to assign distinct memorization biases to different nonlinear regions of the input space. In section 6, we will see how B2S6 improves upon this.

To illustrate the inductive bias of S6, we consider a synthetic task with inputs sampled from

$$\mathbf{u} = (\mathbf{u}_1, \mathbf{0}, \ldots, \mathbf{0}, \mathbf{u}_L), \qquad \mathbf{u}_1 = [u_1, \ldots, u_1]^\top \in \mathbb{R}^d, \quad u_1 \sim \mathcal{N}(0, \sigma_1), \quad \mathbf{u}_L \sim \mathcal{N}(\mathbf{0}, \sigma_2 \mathbf{I}_d),$$

where $\sigma_1, \sigma_2 > 0$ are fixed. The task is to learn the mapping $G(\mathbf{u}) = |u_1|$. Since the target size scales with $\sigma_1$, we report the relative error $|G(\mathbf{u}) - \tilde{G}(\mathbf{u})|/\sigma_1$ to make comparisons fair across different settings. Figure 3 shows the relative errors of different models under different values of $\sigma_1$ and $\sigma_2$. While the S4D model remains robust across all cases, the S6 model fails significantly when $\sigma_1$ is small and $\sigma_2$ is large. This corroborates Theorem 2: a large input $\mathbf{u}_L$ in the half-space $\{\mathbf{u} \mid \mathbf{w}^\top \mathbf{u} > 0\}$ triggers fast forgetting, erasing the memory of $\mathbf{u}_1$ and making the prediction unreliable.

## 5 Mambas are not stable to train

Our discussion so far has focused on the expressiveness and generalization of S6 models. In general, a model lacking expressiveness cannot achieve low training error on complex tasks, while poor generalization refers to models that fit the training data well but perform poorly on test data. If you have trained S6 models on LRA tasks, then you might observe another issue: the training loss curve does not decrease smoothly. Occasionally, a single optimization step causes the model to collapse, turning a high-accuracy model into one that performs no better than random guessing. This reflects a training stability issue rather than one of expressiveness or generalization.

This phenomenon has been empirically studied in the context of general neural networks [21, 19], where instability is often linked to sharp curvature in the loss landscape. In this section, we provide

a theoretical explanation for the training instability observed in S6 models. Since we are not tied to a specific task or loss function, we instead study how the output $\Gamma(\mathbf{u}; \boldsymbol{\Theta})$ depends on the model parameters $\boldsymbol{\Theta}$. While curvature involves second-order derivatives, we focus on first-order gradients. If $(\partial/\partial\boldsymbol{\Theta})\Gamma(\mathbf{u}; \boldsymbol{\Theta})$ is large in magnitude, large curvatures in the loss landscape might appear when we compose $\Gamma(\mathbf{u}; \boldsymbol{\Theta})$ with other functions to form a loss function.

We find that instability in S6 models is closely tied to how the sampling interval $\Delta$ is computed. To formalize this, we prove a theorem comparing the gradients of S6 and S4D models with respect to their $\Delta$-related parameters. For simplicity, we restrict our analysis to a single-input setting, i.e., $d = 1$, and drop all superscripts. Hence, the $\Delta^{(1)}$ for S4D in eq. (2) only depends on a parameter $b = b^{(1)} \in \mathbb{R}$ and the $\Delta_k^{(1)}$ for S6 in eq. (4) only depends on two parameters $w \in \mathbb{R}$ and $b = b^{(1)} \in \mathbb{R}$. Since the specific input distribution is unknown, we make a generic assumption that the inputs are random variables with mild regularity conditions, ensuring that the analysis is broadly applicable.

**Theorem 3.** Assume that $d = 1$ and $n \geq 1$. Let $\Gamma_{\text{S4D}}(\mathbf{u}; \boldsymbol{\Theta})$ and $\Gamma_{\text{S6}}(\mathbf{u}; \boldsymbol{\Theta})$ be given in eq. (1) and (3), respectively, where $\boldsymbol{\Theta}$ is the collection of all model parameters. Given a diagonal matrix $\mathbf{A} \in \mathbb{R}^{n \times n}$ whose diagonal entries are negative, and $\mathbf{B} \in \mathbb{R}^{n \times 1}$, the following statements hold:

1. An S6 model is less stable to train than an S4D model as the input magnitude increases. That is, fix some $L \geq 1$, $b \in \mathbb{R}$, and $w = 0$. Let $\mathbf{u} = (u_1, \ldots, u_L)$ be sampled from a distribution $\mathbb{D}$. For every $\mathbf{C} \in \mathbb{R}^{1 \times n}$, if $\mathbb{E}_{\mathbf{u} \sim \mathbb{D}_L} |(\partial/\partial w)\Gamma_{\text{S6}}|$, $\mathbb{E}_{\mathbf{u} \sim \mathbb{D}_L} |(\partial/\partial b)\Gamma_{\text{S6}}|$, $\mathbb{E}_{\mathbf{u} \sim \mathbb{D}_L} |(\partial/\partial b)\Gamma_{\text{S4D}}| \neq 0$, then

$$\frac{\mathbb{E}_{\mathbf{u} \sim c\mathbb{D}_L} |(\partial/\partial w)\Gamma_{\text{S6}}(\mathbf{u})_L|}{\mathbb{E}_{\mathbf{u} \sim c\mathbb{D}_L} |(\partial/\partial b)\Gamma_{\text{S4D}}(\mathbf{u})_L|} = \Omega(c^3), \qquad \frac{\mathbb{E}_{\mathbf{u} \sim c\mathbb{D}_L} |(\partial/\partial b)\Gamma_{\text{S6}}(\mathbf{u})_L|}{\mathbb{E}_{\mathbf{u} \sim c\mathbb{D}_L} |(\partial/\partial b)\Gamma_{\text{S4D}}(\mathbf{u})_L|} = \Omega(c^2), \qquad c \to \infty.$$

2. An S6 model is less stable to train than an S4D model as the input length increases. That is, fix $w = 0$. There exists a sequence $b(L) \to -\infty$ as $L \to \infty$, such that given any sequence of distributions $\mathbb{D}_L$ on $[0, 1]^{L-1} \times \{1\}$ with $\mathbb{E}_{\mathbf{u} \sim \mathbb{D}_L}[\Gamma_{\text{S4D}}(\mathbf{u})_L] = 0$, $c_1 \leq \text{Var}_{\mathbf{u} \sim \mathbb{D}_L}[u_j] \leq c_2$ for all $1 \leq j \leq L - 1$, where $c_1, c_2 > 0$ are universal constants, and $\left| \sum_{1 \leq i < j \leq L} \text{Cov}(\overline{\mathbf{A}}^{L-i} u_i, \overline{\mathbf{A}}^{L-j} u_j) \right| = \mathcal{O}(L)$, where $\overline{\mathbf{A}} = \exp(\exp(b(L))\mathbf{A})$, we have for a.e. $\mathbf{C} \in \mathbb{R}^{1 \times n}$ that

$$\limsup_{L \to \infty} \left( \frac{\mathbb{E}_{\mathbf{u} \sim \mathbb{D}_L} |(\partial/\partial b(L))\Gamma_{\text{S6}}(\mathbf{u})_L|}{\mathbb{E}_{\mathbf{u} \sim \mathbb{D}_L} |(\partial/\partial b(L))\Gamma_{\text{S4D}}(\mathbf{u})_L|} \right) \Big/ \sqrt{L} > 0.$$

The first part of our result shows that an S6 model becomes increasingly unstable relative to an S4D model as the input size grows. This is expected, as the matrices $\overline{\mathbf{B}}_k^{(i)}$ and $\overline{\mathbf{C}}_k$ in eq. (3) depend linearly on the input $\mathbf{u}_k$. However, this effect is typically mitigated in practice by input normalization and is therefore of less practical concern. The more significant insight lies in the second part of the theorem: as the sequence length $L$ increases, the S6 model also becomes less stable to train. This is particularly relevant in the context of LRA tasks, where sequences can be extremely long; for example, the `PathX` task involves sequences of length 16384. In such cases, the training stability of S6 and S4D are dramatically different. Note that we have made some technical assumptions. First, to preserve long-range dependencies as $L \to \infty$, the sampling interval $\Delta$ must shrink. This justifies why we assumed $b(L) \to -\infty$ so that $\Delta \to 0^+$ as $L \to \infty$. Second, since the pairwise covariances between $\overline{\mathbf{A}}^{L-i} u_i$ and $\overline{\mathbf{A}}^{L-j} u_j$ can be both positive and negative, cancellation occurs and assuming their total sum scales as $\mathcal{O}(L)$ is mild and reflects the natural behavior of aggregated dependencies in long sequences. We also fixed $w = 0$, leaving the case when $w \neq 0$ for future work. More numerical and empirical experiments that illustrate Theorem 3 can be found in Appendix E.

## 6 $\mathbf{B_2S_6}$: an expressive, gently biased, and stable selective model

We introduce $\text{B}_2\text{S}_6$ (Block-Biased-S6) to address the shortcomings of S6. Given an input sequence $\mathbf{u} = (\mathbf{u}_1, \ldots, \mathbf{u}_L)$, where $\mathbf{u}_k \in \mathbb{R}^{d \times 1}$ for $1 \leq k \leq L$, we assume that $d = h \times p$ for two hyperparameters $h, p \geq 1$. We then partition each input vector into $h$ sub-vectors of size $p$: $\mathbf{u}_k = \begin{bmatrix} {}^1\mathbf{u}_k^\top & \cdots & {}^h\mathbf{u}_k^\top \end{bmatrix}^\top$. Note that we use a left-superscript for the block index. Then, the output of a $\text{B}_2\text{S}_6$ unit is a sequence $\Gamma_{\text{B}_2\text{S}_6}(\mathbf{u}) = \mathbf{y} = (\mathbf{y}_1, \ldots, \mathbf{y}_L)$, where each output vector can also be partitioned into $h$ sub-vectors of size $p$, i.e., $\mathbf{y}_k = \begin{bmatrix} {}^1\mathbf{y}_k^\top & \cdots & {}^h\mathbf{y}_k^\top \end{bmatrix}^\top$, defined by

$$ {}^j\mathbf{x}_k^{(i)} = {}^j\overline{\mathbf{A}}_k^{(i)} \, {}^j\mathbf{x}_{k-1}^{(i)} + {}^j\overline{\mathbf{B}}_k^{(i)} \, {}^j\mathbf{u}_k^{(i)}, \quad {}^j\mathbf{y}_k^{(i)} = {}^j\overline{\mathbf{C}}_k \, {}^j\mathbf{x}_k^{(i)}, \qquad 1 \leq i \leq p, \qquad 1 \leq j \leq h, \quad (7)$$

where $^j\mathbf{x}_k^{(i)} \in \mathbb{R}^n$, $^j\mathbf{u}_k^{(i)} \in \mathbb{R}$, $^j\mathbf{y}_k^{(i)} \in \mathbb{R}$, and the matrices $^j\overline{\mathbf{A}}_k^{(i)} \in \mathbb{R}^{n \times n}$, $^j\overline{\mathbf{B}}_k^{(i)} \in \mathbb{R}^{n \times 1}$, and $^j\overline{\mathbf{C}}_k \in \mathbb{R}^{1 \times n}$ are input-dependent and computed at each step $k$ as

$$
\begin{aligned}
^j\overline{\mathbf{A}}_k^{(i)} &= \exp(^j\Delta_k^{(i)}\mathbf{A}), \quad ^j\Delta_k^{(i)} = \mathrm{softplus}(^j\mathbf{w}^\top {}^j\mathbf{u}_k + {}^jb^{(i)}), \\
^j\overline{\mathbf{B}}_k^{(i)} &= \mathbf{A}^{-1}(^j\overline{\mathbf{A}}_k^{(i)} - \mathbf{I})(^j\mathbf{B}_{\mathrm{weight}}\,^j\mathbf{u}_k + {}^j\mathbf{B}_{\mathrm{bias}}^{(i)}), \quad \overline{\mathbf{C}}_k = {}^j\mathbf{u}_k^\top{}^j\mathbf{C},
\end{aligned}
\qquad 1 \le k \le L. \qquad (8)
$$

The trainable parameters are $\mathbf{A} \in \mathbb{R}^{n \times n}$, $^j\mathbf{B}_{\mathrm{weight}} \in \mathbb{R}^{n \times p}$, $^j\mathbf{B}_{\mathrm{bias}}^{(i)} \in \mathbb{R}^{n \times 1}$, $^j\mathbf{C} \in \mathbb{R}^{p \times n}$, $^j\mathbf{w} \in \mathbb{R}^{p \times 1}$, and $^jb^{(i)} \in \mathbb{R}$, for every $1 \le i \le p$ and $1 \le j \le h$. The pseudocode for $B_2S_6$ is given in Algorithm 2, which compares to Algorithm 1 for S6 found in the Mamba paper [25], where $s_\Delta(\cdot) = \mathrm{Broadcast}_d(\mathrm{Linear}_1(\cdot))$ and $^js_\Delta(\cdot) = \mathrm{Broadcast}_p(\mathrm{Linear}_1(\cdot))$ for every $1 \le j \le h$.

| **Algorithm 1** S6 Forward Pass | **Algorithm 2** $B_2S_6$ Forward Pass |
|---|---|
| **Input:** $\mathbf{x} : (B\ L\ d)$ 
 **Output:** $\mathbf{y} : (B\ L\ d)$ | **Input:** $\mathbf{x} : (B\ L\ d)$ 
 **Output:** $\mathbf{y} : (B\ L\ d)$ |
| | 1: **parfor** $j = 1 : h$ **do** ▷ independent for every block |
| | 2: $\quad {}^jI \leftarrow (jp - p + 1) : jp$ ▷ block index |
| 1: $\mathbf{B}_{S6} : (B\ L\ n),\quad \mathbf{B}_{S6}(i,j,:) \leftarrow \mathbf{B}\mathbf{x}(i,j,:)$ | 3: $\quad {}^j\mathbf{B} : (B\ L\ n),\quad {}^j\mathbf{B}(i,j,:) \leftarrow {}^j\mathbf{B}_{\mathrm{weight}}\mathbf{x}(i,j,{}^jI)$ |
| | 4: $\quad {}^j\mathbf{B} : (B\ L\ p\ n) \leftarrow \mathrm{Broadcast}_p({}^j\mathbf{B}) + \mathrm{Broadcast}_L({}^j\mathbf{B}_{\mathrm{bias}})$ |
| 2: $\mathbf{C}_{S6} : (B\ L\ n),\quad \mathbf{C}_{S6}(i,j,:) \leftarrow \mathbf{x}(i,j,:)^\top\mathbf{C}$ | 5: $\quad {}^j\mathbf{C} : (B\ L\ n),\quad {}^j\mathbf{C}(i,j,:) \leftarrow \mathbf{x}(i,j,{}^jI)^\top{}^j\mathbf{C}$ |
| 3: $\Delta : (B\ L\ d) \leftarrow \mathrm{softplus}(s_\Delta(\mathbf{x}) + \mathbf{b})$ | 6: $\quad {}^j\Delta : (B\ L\ p) \leftarrow \mathrm{softplus}({}^js_\Delta(\mathbf{x}(:,:,{}^jI)) + {}^j\mathbf{b})$ |
| 4: $\overline{\mathbf{A}}, \overline{\mathbf{B}} : (B\ L\ d\ n) \leftarrow \mathrm{discretize}(\Delta, \mathbf{A}, \mathbf{B}_{S6})$ | 7: $\quad {}^j\overline{\mathbf{A}}, {}^j\overline{\mathbf{B}} : (B\ L\ p\ n) \leftarrow \mathrm{discretize}({}^j\Delta, \mathbf{A}, {}^j\mathbf{B})$ |
| 5: $\mathbf{y} \leftarrow \mathrm{SSM}(\overline{\mathbf{A}}, \overline{\mathbf{B}}, \mathbf{C}_{S6})(\mathbf{x})$ | 8: $\quad \mathbf{y}(:,:,{}^jI) \leftarrow \mathrm{SSM}({}^j\overline{\mathbf{A}}, {}^j\overline{\mathbf{B}}, {}^j\mathbf{C})(\mathbf{x}(:,:,{}^jI))$ |
| | 9: **end parfor** |

Comparing the $B_2S_6$ unit in eq. (7) to the S6 unit in eq. (3), we highlight two key differences (see Figure 1). First, $B_2S_6$ partitions the $d$-dimensional input into $h$ blocks of $p$-dimensional sub-vectors and applies an independent recurrent unit to each block. Second, we introduce a bias term $^j\mathbf{B}_{\mathrm{bias}}^{(i)}$ in the computation of the input matrix $^j\overline{\mathbf{B}}_k$. While this term is input-independent, it varies across channels, increasing the model's effective width. We do not claim that we invented the block unit design, which was previously suggested in the Mamba2 paper [20], but in this work, we rigorously demonstrate how the combination of block structure and channel-specific bias enhances the expressiveness and generalization of the model on long-range sequence modeling tasks.

First, we show that unlike an S6 model, a $B_2S_6$ model regains the universal approximation property studied in Theorem 1 by introducing *either* the block unit *or* the bias unit.

**Theorem 4.** Fix a constant for $^j\Delta_k^{(i)}$ for all $i, j$, and $k$ in eq. (7). The following two statements hold:

1. The block unit alone makes $B_2S_6$ a universal approximator. More precisely, set $^j\mathbf{B}_{\mathrm{bias}}^{(i)} = \mathbf{0}$ for all $i$ and $j$, and let $\sigma : \mathbb{R} \to \mathbb{R}$ be any Lipschitz continuous, non-polynomial activation function. Given any continuous function $G : [0,1]^{L-1} \times \{1\} \to \mathbb{R}$ and any $\epsilon > 0$. There exist some $h$ and $p$ with $h \times p = d \ge 1$, $n \ge 1$, and a choice of parameters $\mathbf{M}, \mathbf{N}, \boldsymbol{\theta}, \mathbf{A}, {}^j\mathbf{B}_{\mathrm{weight}}$, and $^j\mathbf{C}$, where $1 \le j \le h$, such that the map $\tilde{G}$ in eq. (5) with $\Gamma = \Gamma_{B_2S_6}$ in eq. (7) satisfies that

$$|\tilde{G}(\mathbf{u}) - G(\mathbf{u})| \le \epsilon, \qquad \text{for any } \mathbf{u} \in [0,1]^{L-1} \times \{1\}.$$

2. The bias unit alone makes $B_2S_6$ a universal approximator. More precisely, set $h = 1$ and $p = d$, and let $\sigma : \mathbb{R} \to \mathbb{R}$ be any Lipschitz continuous, non-polynomial activation function. Given any continuous function $G : [0,1]^{L-1} \times \{1\} \to \mathbb{R}$ and any $\epsilon > 0$. There exist some $d \ge 1$, $n \ge 1$, and a choice of parameters $\mathbf{M}, \mathbf{N}, \boldsymbol{\theta}, \mathbf{A}, {}^1\mathbf{B}_{\mathrm{weight}}, {}^1\mathbf{B}_{\mathrm{bias}}^{(i)}$, and $^1\mathbf{C}$, where $1 \le i \le d$, such that the map $\tilde{G}$ in eq. (5) with $\Gamma = \Gamma_{B_2S_6}$ in eq. (7) satisfies that

$$|\tilde{G}(\mathbf{u}) - G(\mathbf{u})| \le \epsilon, \qquad \text{for any } \mathbf{u} \in [0,1]^{L-1} \times \{1\}.$$

The block design and channel-specific bias in the $B_2S_6$ model significantly enhance its expressiveness, enabling it to handle more complex sequential tasks that benefit from wider neural networks. To illustrate this, we revisit the experiment from section 3, this time using only the block structure

(without the bias term), noting that the addition of the bias would only further improve performance. As shown in Figure 2, unlike an S6 model, the performance of the $B_2S_6$ model improves substantially as $d$ increases, holding the ratio $h/p$ constant.

In contrast to the S6 model, the $B_2S_6$ architecture introduces a gentler inductive bias that is better suited for long-range tasks. Instead of Theorem 2, we now establish the following result.

**Theorem 5.** Fix a $k_0 < L$ and let $\mathbf{A} \in \mathbb{R}^{n \times n}$ be a diagonal matrix with negative diagonal entries. For a.e. input sequence $\mathbf{u} = (\mathbf{u}_1, \ldots, \mathbf{u}_L) \in \mathbb{R}^L$ such that there exist $j_1$ and $j_2$ with ${}^{j_1}\mathbf{w}^\top {}^{j_1}\mathbf{u}_{k_0} > 0$ and ${}^{j_2}\mathbf{w}^\top {}^{j_2}\mathbf{u}_{k_0} < 0$, and a.e. parameters ${}^j\mathbf{B}_{\mathrm{weight}}$, ${}^j\mathbf{B}_{\mathrm{bias}}^{(i)}$, ${}^j\mathbf{C}$, and ${}^jb^{(i)}$, where $1 \leq j \leq h$ and $1 \leq i \leq p$, let $S_{B_2S_6,k}$ be defined in eq. (6), where $\Gamma = \Gamma_{B_2S_6}$ is defined in eq. (7). We have

$$
S_{B_2S_6,k}((\mathbf{u}_1, \ldots, \mathbf{u}_{k_0-1}, c\mathbf{u}_{k_0}, \mathbf{u}_{k_0+1}, \ldots, \mathbf{u}_L)) = \begin{cases} \mathcal{O}(|c|^{-2}) & , k < k_0, \\ \mathcal{O}(|c|^{-1}) & , k = k_0, \qquad c \to \pm\infty. \\ \Theta(1) & , k > k_0, \end{cases}
$$

If each ${}^j\mathbf{w}$ is randomly initialized for $1 \leq j \leq h$, then the probabilities that a given input satisfies ${}^j\mathbf{w}^\top {}^j\mathbf{u}_{k_0} < 0$ and ${}^j\mathbf{w}^\top {}^j\mathbf{u}_{k_0} > 0$ are equal. As a result, the probability that the assumptions in Theorem 5 are violated vanishes exponentially with the number of blocks $h$. Compared to Theorem 2, this means that $B_2S_6$ exhibits a significantly milder inductive bias in the presence of large inputs, avoiding exponentially decaying effects as the input magnitude increases. This behavior is confirmed by the experiment in section 4, where $B_2S_6$ maintains strong performance even when $\sigma_1$ is very small and $\sigma_2$ is very large. Lastly, while the block and bias components improve expressiveness and inductive bias, they do not directly resolve the training instability that arises with long sequences. Therefore, for LRA tasks, we reduce the learning rates of ${}^j\mathbf{w}$ and ${}^jb^{(i)}$ to improve training stability.[3]

# 7 Experiments

**Ablation.** In this paper, we show that the $B_2S_6$ model achieves strong performance on the LRA benchmark. Compared to the standard Mamba model, our $B_2S_6$ variant incorporates a multihead structure and a bias term $\mathbf{B}_{\mathrm{bias}}$, and uses complex-valued parameters for $\mathrm{diag}(\mathbf{A})$, $\mathbf{B}_{\mathrm{weight}}$, and $\mathbf{B}_{\mathrm{bias}}$ (see [64, 97, 47]). Before presenting full LRA results, we demonstrate the utility of each modification in Table 2. The second column shows model accuracy without bias terms, the third without complex parameterization, and all rows below the first use the multihead structure. As seen in Table 2, all three components contribute positively to the performance on the sCIFAR-10 task. An interesting observation is that, when $d = h \times p$ is fixed, model accuracy increases monotonically with $h$ only in the absence of $\mathbf{B}_{\mathrm{bias}}$. This should not be surprising: if $\mathbf{B}_{\mathrm{bias}} = \mathbf{0}$, the model's effective width is $h$, so larger $h$ improves expressiveness and generalization. However, once $\mathbf{B}_{\mathrm{bias}}$ is introduced, the model already attains high effective (though non-selective) width even when $h = 1$ because ${}^j\mathbf{B}_{\mathrm{bias}}^{(i)}$ varies with $i$. Thus, increasing $h$ trades off between more selective blocks, which improves the *quantity* of selectivity, and reduced receptive field ${}^j\mathbf{u}_k$ per block, which impairs the *quality* of each selection.

**Long-Range Arena.** In Table 3, we see that our $B_2S_6$ model outperforms many selective and non-selective models on LRA. In particular, it is the first selective SSM we are aware of that achieves state-of-the-art performance on this benchmark. A more comprehensive table is found in Appendix F.

**Language Tasks.** While our primary goal is not to train a large language model (LLM), we conduct a preliminary evaluation of $B_2S_6$'s language modeling capability using a sampled version of the SlimPajama-627B dataset [78]. For each model family, we train a model with approximately 30 layers, corresponding to approximately 250M parameters. The precise number of layers varies slightly with models to make the number of parameters roughly the same. We train four models for one epoch and report the perplexity statistics. We only vary the recurrent unit without changing the rest of the model architecture. That is, all models are based on the Mamba architecture proposed in [25, Fig. 3]:

$$
\mathbf{y} = \Gamma(\sigma(\mathrm{Conv}(\mathrm{Linear}(\mathbf{x})))) \circ \sigma(\mathrm{Linear}(\mathbf{x})),
$$

---

[3]In general, freezing some of the other parameters indeed helps stabilize the training, but it also puts us at risk of converging to bad minima. For example, we observed that it never works to freeze $\mathbf{B}$ and $\mathbf{C}$, and that freezing the matrix $\mathbf{A}$ and $\mathbf{D}$ usually does not have a big impact, given that they are initialized properly (see [26, 97]).

Table 2: Ablation study of our $B_2S_6$ model. We train a model to learn the sCIFAR-10 task, where we fix $d = 128$ and change the number of blocks $h$ and the size of each block $p$ (see eq. (7)). For each pair of $h$ and $p$, we train a full $B_2S_6$ model with complex parameters, a $B_2S_6$ model with complex parameters but no bias term, and a full $B_2S_6$ model with real parameters. A more greenish color indicates a better accuracy, while a more reddish color labels a worse accuracy.

| | | With $B_{bias}$ (Complex) | Without $B_{bias}$ (Complex) | With $B_{bias}$ (Real) |
|---|---|---|---|---|
| $h$ | $p$ | Accuracy $\pm$ Std | Accuracy $\pm$ Std | Accuracy $\pm$ Std |
| 1 | 128 | **81.56** $\pm$ 1.22 | **71.77** $\pm$ 1.88 | **47.82** $\pm$ 5.42 |
| 2 | 64 | **84.83** $\pm$ 0.73 | **79.44** $\pm$ 1.10 | **54.40** $\pm$ 0.35 |
| 4 | 32 | **83.86** $\pm$ 1.38 | **80.93** $\pm$ 0.94 | **54.77** $\pm$ 0.54 |
| 8 | 16 | **87.19** $\pm$ 0.26 | **81.54** $\pm$ 1.14 | **53.81** $\pm$ 0.87 |
| 16 | 8 | **87.04** $\pm$ 0.33 | **83.78** $\pm$ 0.79 | **51.79** $\pm$ 3.61 |
| 32 | 4 | **85.83** $\pm$ 0.80 | **84.11** $\pm$ 0.39 | **56.18** $\pm$ 0.18 |
| 64 | 2 | **86.30** $\pm$ 0.60 | **84.27** $\pm$ 0.76 | **52.03** $\pm$ 2.04 |
| 128 | 1 | **85.32** $\pm$ 0.55 | **84.79** $\pm$ 0.43 | **56.45** $\pm$ 0.71 |

Table 3: Test accuracies in the Long-Range Arena of different variants of SSMs. A bold number indicates the best accuracy on a task while an underlined number corresponds to the second best.

| Model | ListOps | Text | Retrieval | Image | Pathfinder | Path-X | Avg. |
|---|---|---|---|---|---|---|---|
| S4 [28] | 59.60 | 86.82 | 90.90 | 88.65 | 94.20 | 96.35 | 86.09 |
| S4D [27] | 60.47 | 86.18 | 89.46 | 88.19 | 93.06 | 91.95 | 84.89 |
| S5 [77] | 62.15 | **89.31** | 91.40 | 88.00 | 95.33 | **98.58** | 87.46 |
| S6 (Mamba) [25] | 38.02 | 82.98 | 72.14 | 69.82 | 69.26 | 67.32 | 66.59 |
| S7 [79] | 63.77 | 87.22 | **91.80** | 61.14 | 65.62 | 61.50 | 71.82 |
| Mamba2 [20] | 41.45 | 86.09 | 79.23 | 71.96 | 75.45 | 69.07 | 70.54 |
| $B_2S_6$ (ours) | **63.85** 
 ±0.45 | 88.32 
 ±0.50 | 91.44 
 ±0.36 | **88.81** 
 ±0.22 | **95.93** 
 ±0.38 | 97.90 
 ±0.17 | **87.71** |

where $\circ$ stands for the Hadamard product and we only change $\Gamma$ in three models to be $\Gamma_{S4D}, \Gamma_{S6}$, $\Gamma_{Mamba2}$, and $\Gamma_{B_2S_6}$, respectively. We find that introducing the bias term $\mathbf{B}_{bias}$ only makes the training of a $B_2S_6$ model $2.7\%$ slower than an S6 model of comparable size. Yet, the perplexity of $B_2S_6$ closely matches that of S6, showing the versatility of our model.

Figure 4: Perplexity over training steps for an S6, Mamba2, $B_2S_6$, and S4D model. The dataset is SlimPajama-6B, a sampled version of the SlimPajama-627B dataset.

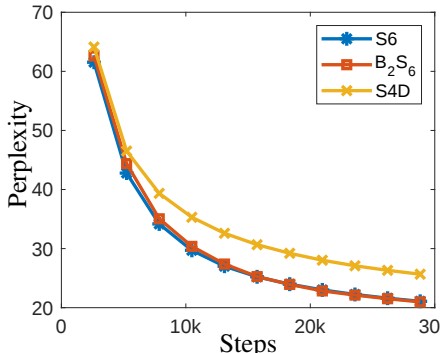

| Steps | S6 | Mamba2 | $B_2S_6$ | S4D |
|---|---|---|---|---|
| 2620 | 61.5441 | 59.6233 | 62.6133 | 64.1203 |
| 5241 | 42.7838 | 39.8305 | 44.3846 | 46.5118 |
| 7862 | 34.1740 | 31.9994 | 35.0440 | 39.3496 |
| 10483 | 29.7074 | 27.0987 | 30.3787 | 35.3028 |
| 13104 | 27.0171 | 25.2930 | 27.3891 | 32.5946 |
| 15725 | 25.2265 | 24.0179 | 25.2668 | 30.6639 |
| 18346 | 23.9789 | 23.1224 | 23.9370 | 29.2052 |
| 20967 | 23.0044 | 22.3695 | 22.8394 | 28.0314 |
| 23588 | 22.2368 | 21.7509 | 22.1322 | 27.0897 |
| 26209 | 21.6068 | 21.2480 | 21.5206 | 26.3221 |
| 28830 | 21.0785 | 20.8218 | 20.9773 | 25.6661 |

# 8   Conclusion

In this work, we provided a theoretical analysis of Mamba's limitations in modeling long-range sequences, focusing on expressiveness, inductive bias, and training stability. We proposed a new model, $B_2S_6$, which introduces a block structure and channel-specific bias to improve upon each of these dimensions. Empirical results demonstrate that $B_2S_6$ significantly enhances performance on long-range sequence tasks while maintaining versatility across domains. Future work includes a more detailed analysis of training stability across all Mamba parameters, exploring architectural refinements to further mitigate instability, studying the training dynamics of Mamba and $B_2S_6$ to better characterize their inductive biases, and scaling up $B_2S_6$ as a language or foundation model.

**Acknowledgments**

AY was partially supported by the Office of Naval Research under Grant Number N00014-23-1-2729 and NSF DMS-2319621. NBE would like to acknowledge the U.S. Department of Energy, under Contract Number DE-AC02-05CH11231 and DE-AC02-05CH11231, for providing partial support of this work.

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

# A Related works

**Sequence Models.** Sequential data appear across a wide range of domains, including natural language processing, computer vision, generative modeling, and scientific machine learning. To model such data, primitive deep learning approaches primarily relied on recurrent neural networks (RNNs) and their variants [6, 13, 22, 68, 55], as well as convolutional neural networks (CNNs) [9, 66]. The last decade has witnessed the dominance of the transformer architecture [37, 17, 41, 106, 54], following the seminal work [83]. Recently, a new class of models known as state-space models (SSMs) has emerged as a promising competitor [28, 27, 30, 77, 25, 20]. These models represent sequences through an underlying continuous-time dynamical system and offer a key advantage: they can handle sequences of varying lengths with a fixed number of parameters [85] and be trained with time and memory complexity that scales linearly with sequence length [28, 77, 25].

**State Space Models.** The term "state space models" (SSMs) originates from control theory and dates back to [36]. The first widely adopted SSM in machine learning was the S4 model proposed by [28]. Both the original S4 and its successor, Liquid-S4 [30], use a diagonal-plus-rank-one (DPRO) structure in the state matrix to significantly reduce computational cost compared to traditional RNNs. Later, [27] showed that a purely diagonal state matrix could achieve comparable performance, leading to the simplified S4D model. Since then, most SSMs have adopted diagonal state matrices, including S5 [77], Regularized SSM [46], Stable-SSM [86], S4-PTD, and S5-PTD [99, 63]. While a diagonal parameterization explores the structural simplicity of SSMs, [59] analyzes the computational acceleration of LTI systems and [12] investigates its numerical stability. Other works explore implicit sequence models that do not rely directly on an LTI formulation or rely on a modified LTI formulation, such as [98, 1, 56, 69]. An extensive list of different SSM architectures has been given in the survey articles [89, 58]. The expressiveness of SSMs has been studied in [98, 47], their training dynamics in [76, 97], and generalization properties in [97, 46]. Several studies have also investigated the representation stability of SSMs, including [86, 98]. Applications of SSMs in different scientific fields have been studied in several papers [102, 90, 63].

**Mamba Models.** Mamba [25] extends SSMs by introducing input-dependent dynamics via a selective mechanism, resulting in a highly efficient sequential architecture. Unlike earlier SSMs that use fixed, input-independent recurrence, Mamba adapts its dynamics at each step, enabling it to rival transformer-based models in language tasks. The theoretical benefits of this input-dependent recurrence was studied in [53] through the lens of controlled differential equations (CDEs). Mamba has since inspired a number of extensions and studies. Mamba2 [20] incorporates a multihead structure with softmax gating. Mamba has also been applied beyond language modeling [58], including computer vision [107, 103, 44], time-series forecasting [91], multimodal learning [62], and audio processing [43]. In addition, [51] proposed using Mamba as a foundation model backbone across modalities. Despite its versatility, follow-up studies such as [3] have highlighted the limitations of Mamba on long-range sequence benchmarks such as LRA. The S7 model [79] has been proposed as another selective model with slightly improved performances on LRA tasks; though, the accuracies are still substantially worse than models like S4D and S5. Other efforts to enhance Mamba's long memory retention and selectivity mechanism are found in [95] and [65], respectively. The spectral properties of the state matrix are considered in [24] and related to Mamba's state-tracking capabilities. In addition, the recent work [32] proposes a theoretical generalization upper bound based on the Rademacher complexity of Mamba models. Notably, [74] conducted a thorough comparison of these different recurrent units, including S4, Mamba, and traditional RNNs. Many works have shown the advantages of a bidirectional structure in SSMs and Mambas [34, 48], which we also adopt in training the LRA benchmark tasks.

**Universal Approximation Theorems.** In this paper, we use the universal approximation theorem (UAT) as a tool to assess a model's expressiveness. UATs ask whether a target function can be approximated to arbitrary accuracy by a sufficiently large neural network. This line of work dates back to [18], who proved that two-layer neural networks with sigmoid activation are universal approximators. This result was later extended to networks with any continuous, non-polynomial activation function. While many classical results focus on shallow but wide networks, recent work by [39] established that deep, narrow networks can also be universal approximators. Universal approximation properties have been studied across various architectures. Notable examples include shallow and wide operator neural networks [50, 15], as well as their deep and narrow counterparts [96]. For sequence models, UATs are known for RNNs [71], CNNs [105], Transformers [101], and

SSMs [86]. There are limited works on the universal approximation properties of Mamba models. One notable exception is [53], where the uniform closure of Mamba (i.e., functions that can be uniformly approximated by Mamba) is analyzed in a continuous setting (i.e., the inputs are continuous time series instead of discrete sequences) through linear controlled differential equations. The analysis there highlights the important role of an input-dependent sampling interval $\Delta$ in the expressiveness of Mamba, suggesting that Theorem 1 may not hold for a trainable $\Delta$ (see Appendix G).

**Width and Depth of Neural Networks.** While UATs focus on the density of neural networks in a given topology, more practically relevant questions involve the rate of approximation, e.g., how deep or wide a network must be to approximate a target function within a given accuracy. One of the most celebrated results is by [82], which shows that depth improves expressiveness more efficiently than width. However, in practice, deep but narrow networks often face optimization challenges during training [23, 57, 16, 67]. In contrast, networks with greater effective width tend to enjoy better theoretical guarantees on generalization. This advantage can be understood through both the neural tangent kernel (NTK) perspective [35, 7, 2, 11, 100] and mean-field theory [52, 80, 10], both of which highlight the benefits of wide models in training dynamics and generalization performance.

**Sensitivity Analysis.** Neural networks often operate on high-dimensional inputs, and a large body of work has studied how to quantify the sensitivity of the output to individual input components — a line of research typically referred to as attribution. While our paper focuses on relative gradients, many other gradient-based attribution methods have been developed, including DeepLIFT [73], integrated gradients [81], sensitivity-$n$ [4], and others [8, 75]. Most of these methods compare a given input to a baseline or reference input, whereas our analysis focuses specifically on the effect of large-magnitude inputs in isolation without reference to a benchmark input.

**Training Stability and the Loss Landscape.** Training stability in deep neural networks has been extensively studied through the lens of loss landscape geometry. Sharp minima and high-curvature regions are often associated with poor generalization and unstable optimization [38, 14, 42]. Gradient explosion or vanishing can lead to optimization failure [23, 57] for recurrent neural networks. More recent works have linked curvature and gradient dynamics to the trainability of neural networks [16, 67, 70]. These studies emphasize the importance of smooth, well-conditioned landscapes for stable training, motivating architectural choices [31, 93] and learning rate schedulers [49] that mitigate instability.

## B  Proofs of UAT and non-UAT results

In this section, we prove all technical results related to UATs presented in section 3 and 6. Recall that the univariate neural network architecture that we study in this paper is given by

$$\tilde{G}(\mathbf{u}) = \tilde{G}((u_1, \ldots, u_L)) = \mathbf{N}\sigma(\Gamma((\mathbf{M}u_1, \ldots, \mathbf{M}u_L))_L + \boldsymbol{\theta}).$$

A pictorial illustration of this architecture is given in Figure 5.

### B.1  Proof of Theorem 1

To prove that such a $\tilde{G}$ is not a universal approximator when $\Gamma = \Gamma_{S6}$, we will see that since an S6 model ties the matrices $\overline{\mathbf{B}}_k$ and $\overline{\mathbf{C}}_k$ for all channels, the system becomes a quadratic encoder of the input sequence. We need the following technical lemma to show that a quadratic encoder is not invertible, and therefore, information will be lost when we apply this encoder to an input sequence.

**Lemma 1.** Given any $L \geq 3$, there exists a continuous function $G : [0, 1]^L \rightarrow \mathbb{R}$ such that given any quadratic polynomial of $L$ variables $P : [0, 1]^{L-1} \times \{1\} \rightarrow \mathbb{R}$, there exist two points $\mathbf{x}, \mathbf{y} \in [0, 1]^{L-1} \times \{1\}$ such that

$$P(\mathbf{x}) = P(\mathbf{y}), \qquad |G(\mathbf{x}) - G(\mathbf{y})| > 1.$$

*Proof.* Since the restriction of a quadratic polynomial of $L$ variables to 3 variables is still a quadratic polynomial, without loss of generality, we assume that $L = 3$. Moreover, instead of assuming that $G$ and $P$ are functions on $[0, 1]^{3-1} \times \{1\}$, we can assume that they are functions on $[0, 1]^2$. It is well-known that there is a number $N \geq 1$ such that every bivariate quadratic polynomial has at most $N$ strict local minima or maxima. We construct $G$ by selecting $N + 1$ arbitrary distinct points $\mathbf{x}_1, \mathbf{x}_2, \ldots, \mathbf{x}_{N+1}$ in $(0, 1)^2$. Around each point $\mathbf{x}_j$, we make a small disk $D_j = D_\rho(\mathbf{x}_j)$.

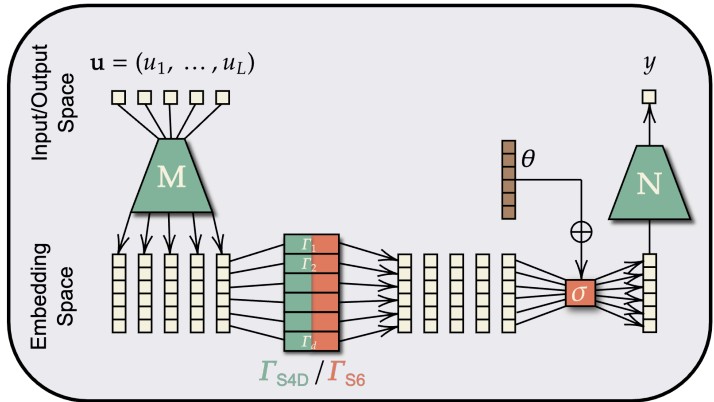

Figure 5: The architecture of the neural network eq. (5) that we study in this paper. In this picture, a horizontal operator is applied channel-wise to every sequence in a channel, and a vertical operation is applied element-wise to every position in a sequence. A green color indicates a linear operator while an orange color indicates a nonlinear one.

By taking $\rho$ small enough, we can also ensure that $D_j \subset [0,1]^2$ for all $1 \leq j \leq N+1$ and that $D_1, D_2, \ldots, D_{N+1}$ are mutually disjoint. Let $G$ be a continuous function such that $G(\mathbf{x}_j) = 0$ for every $1 \leq j \leq N+1$ and $G$ equals 2 on $\partial D_j$, the boundary of $D_j$, for every $1 \leq j \leq N+1$. Such a $G$ clearly exists by Urysohn's lemma. We claim that $G$ satisfies the condition in the our lemma. To see this, given any arbitrary quadratic polynomial $P : [0,1]^2 \to \mathbb{R}$, we let $S_j$ be the connected component of the set $\{\mathbf{x}|P(\mathbf{x}) = P(\mathbf{x}_j)\}$ that contains $\mathbf{x}_j$. There are two possibilities: either $S_j$ intersects $\partial D_j$ or not. If $S_j$ does not intersect $\partial D_j$, then that means there is a strict local minimum or a strict local maximum of $P$ in $D_j$, but $D_1, \ldots, D_{N+1}$ are mutually disjoint and there are at most $N$ local minima or maxima of $P$. Hence, there is at least one $j$ such that $S_j$ intersects $\partial D_j$. Let $\mathbf{y} \in S_j \cap \partial D_j$. Then, we have that

$$P(\mathbf{x}_j) = P(\mathbf{y}), \qquad |G(\mathbf{x}_j) - G(\mathbf{y})| = |0 - 2| > 1.$$

The proof is complete. $\qquad\square$

We are now ready to prove Theorem 1. The proof of the first part is built upon a result from [60] for UAT of a two-layer wide neural network to approximate any continuous function and a result from [88] for UAT of LTI systems to approximate any convolutional kernel.

*Proof of Theorem 1.* We prove the two statements separately.

**Proof of Part I.** Without loss of generality, assume that $\sigma$ is 1-Lipschitz.[4] Let a continuous function $G : [0,1]^L \to \mathbb{R}$ and an error tolerance $\epsilon > 0$ be given. By [60], there exists a function $\tilde{G} : [0,1]^L \to \mathbb{R}$ of the form

$$\tilde{G}(\mathbf{u}) = \sum_{i=1}^{d} f^{(i)} \sigma((\mathbf{w}^{(i)})^\top \mathbf{u} + \theta^{(i)}),$$

where $\mathbf{w}^{(i)} \in \mathbb{R}^L$ and $b_i \in \mathbb{R}$ for every $1 \leq i \leq d$, such that

$$|\tilde{G}(\mathbf{u}) - G(\mathbf{u})| \leq \frac{\epsilon}{2}$$

for every $\mathbf{u} \in [0,1]^L$. For any $1 \leq i \leq d$, by [87], there exist $n_i \geq 1$, $\mathbf{A}^{(i)} \in \mathbb{R}^{n_i \times n_i}$, $\mathbf{B}^{(i)} \in \mathbb{R}^{n_i \times 1}$, and $\mathbf{C}^{(i)} \in \mathbb{R}^{1 \times n_i}$, such that the matrices $\overline{\mathbf{A}}^{(i)}$, $\overline{\mathbf{B}}^{(i)}$, and $\overline{\mathbf{C}}^{(i)}$ from the discretized LTI system (see eq. (2)) satisfy that

$$|(\mathbf{w}_j)^{(i)} - \overline{\mathbf{C}}^{(i)}(\overline{\mathbf{A}}^{(i)})^{L-j}\overline{\mathbf{B}}^{(i)}| < \frac{\epsilon}{2\sqrt{d}L(\|\mathbf{f}\|_2 + 1)}$$

---

[4]Otherwise, if $\sigma$ is $\ell$-Lipschitz, we can divide $\sigma$ by $\ell$ to make it 1-Lipschitz and multiply $\mathbf{N}$ by $\ell$ so that the value of $\tilde{G}$ does not change.

for all $1 \leq j \leq L$, where $\mathbf{f} = [f^{(1)} \cdots f^{(d)}]^\top$. Hence, we have that

$$\left| \sigma((\mathbf{w}^{(i)})^\top \mathbf{u} + \theta^{(i)}) - \sigma\left( \sum_{j=1}^{L} \overline{\mathbf{C}}^{(i)}(\overline{\mathbf{A}}^{(i)})^{L-j}\overline{\mathbf{B}}^{(i)}u_j + \theta^{(i)} \right) \right|$$

$$\leq \left| (\mathbf{w}^{(i)})^\top \mathbf{u} - \left( \sum_{j=1}^{L} \overline{\mathbf{C}}^{(i)}(\tilde{\mathbf{A}}^{(i)})^{L-j}\overline{\mathbf{B}}^{(i)}u_j \right) \right|$$

$$\leq \sqrt{ \sum_{j=1}^{L} \left( (\mathbf{w}^{(i)})_j - \overline{\mathbf{C}}^{(i)}(\overline{\mathbf{A}}^{(i)})^{L-j}\overline{\mathbf{B}}^{(i)} \right)^2 } \, \|\mathbf{u}\|_2 \leq L \frac{\epsilon}{2\sqrt{d}L(\|\mathbf{f}\|_2 + 1)} = \frac{\epsilon}{2\sqrt{d}(\|\mathbf{f}\|_2 + 1)},$$

where the first inequality follows from the Lipschitz continuity of $\sigma$ and the second inequality follows from the Hölder's inequality. Set $n = \sum_{i=1}^{d} n_i$ and define a block-diagonal matrix $\mathbf{A}$ so that

$$\mathbf{A} = \begin{bmatrix} \mathbf{A}^{(1)} & & & \\ & \mathbf{A}^{(2)} & & \\ & & \ddots & \\ & & & \mathbf{A}^{(d)} \end{bmatrix}.$$

For every $1 \leq i \leq d$, define $\mathbf{B}^{(i)} \in \mathbb{R}^{n \times 1}$ to be a sparse column vector so that $\mathbf{B}^{(i)}((\sum_{i'=1}^{i-1} n_{i'} + 1) : (\sum_{i'=1}^{i} n_{i'})) = \mathbf{B}^{(i)}$ and is zero elsewhere. Similarly, define $\mathbf{C}^{(i)} \in \mathbb{R}^{1 \times n}$ to be a sparse column vector so that $\mathbf{C}^{(i)}((\sum_{i'=1}^{i-1} n_{i'} + 1) : (\sum_{i'=1}^{i} n_{i'})) = \mathbf{C}^{(i)}$ and is zero elsewhere. Now, it is easy to see that given these definitions, the S4D defined in eq. (1) satisfies that

$$(\mathbf{\Gamma}_{\text{S4D}}((u_1, \ldots, u_L))_L)^{(i)} = \sum_{j=1}^{L} \overline{\mathbf{C}}^{(i)}(\overline{\mathbf{A}}^{(i)})^{L-j}\overline{\mathbf{B}}^{(i)}u_j$$

for every $1 \leq i \leq d$. Hence, let $\tilde{G}_{\text{S4D}}$ be such that

$$\tilde{G}_{\text{S4D}} : [0,1]^L \to \mathbb{R}, \quad (u_1, \ldots, u_L) \mapsto \mathbf{f}\sigma(\mathbf{\Gamma}_{\text{S4D}}((Mu_1, \ldots, Mu_L))_L + \boldsymbol{\theta}),$$

where $\mathbf{M} = [1 \cdots 1]^\top$. For any $\mathbf{u} \in \mathbb{R}^L$, we have that

$$|\tilde{G}_{\text{S4D}}(\mathbf{u}) - \tilde{G}(\mathbf{u})| \leq \|\mathbf{f}\|_2 \sqrt{ \sum_{i=1}^{d} \left( \sigma((\mathbf{w}^{(i)})^\top \mathbf{u}) - \sigma\left( \sum_{j=1}^{L} \overline{\mathbf{C}}^{(i)}(\overline{\mathbf{A}}^{(i)})^{L-j}\overline{\mathbf{B}}^{(i)}u_j \right) \right)^2 }$$

$$\leq \|\mathbf{f}\|_2 \sqrt{d} \frac{\epsilon}{2\sqrt{d}(\|\mathbf{f}\|_2 + 1)} < \frac{\epsilon}{2}.$$

Now, for any $\mathbf{u} \in \mathbb{R}^L$, we have that

$$|\tilde{G}_{\text{S4D}}(\mathbf{u}) - G(\mathbf{u})| \leq |\tilde{G}_{\text{S4D}}(\mathbf{u}) - \tilde{G}(\mathbf{u})| + |\tilde{G}(\mathbf{u}) - G(\mathbf{u})| \leq \frac{\epsilon}{2} + \frac{\epsilon}{2} \leq \epsilon.$$

The statement is proved.

**Proof of Part II.** Given any sequence $\mathbf{u} \in \mathbb{R}^L$, any encoder $\mathbf{M} = [m^{(1)} \cdots m^{(L)}]^\top$, and any Mamba parameters $\mathbf{A} \in \mathbb{R}^{n \times n}$, $\mathbf{B} \in \mathbb{R}^{n \times d}$, and $\mathbf{C} \in \mathbb{R}^{d \times n}$, we first show how the Mamba system in eq. (3) can be simplified. For any $1 \leq i \leq d$, we have that

$$\mathbf{x}_{k+1}^{(i)} = \overline{\mathbf{A}}\mathbf{x}_k^{(i)} + (\overline{\mathbf{P}}u_k)m^{(i)}u_k,$$
$$(\mathbf{y}_k)^{(i)} = (u_k\overline{\mathbf{Q}})\mathbf{x}_k^{(i)}, \tag{9}$$

where $\overline{\mathbf{A}} = \exp(\mathbf{A})$, $\mathbf{P} = \mathbf{A}^{-1}(\overline{\mathbf{A}} - \mathbf{I})\mathbf{BM}$, and $\overline{\mathbf{Q}} = \mathbf{M}^\top \mathbf{C}$ are fixed matrices depending only on $\mathbf{A}$, $\mathbf{B}$, $\mathbf{C}$, and $\mathbf{M}$. Hence, the $i$th entry of the final output is given by

$$\mathbf{\Gamma}_{\text{S6}}((Mu_1, \ldots, Mu_L))_L^{(i)} = m^{(i)} u_L \underbrace{\left( \sum_{j=1}^{L} \overline{\mathbf{Q}}\,\overline{\mathbf{A}}^{L-j}\,\overline{\mathbf{P}}u_j^2 \right)}_{F((u_1, \ldots, u_L))}.$$

Importantly, note that since $u_L = 1$, $F$ is a quadratic function in $u_1, \ldots, u_L$ that does not depend on $i$. The output of $\tilde{G}_{\text{S6}}$ can then be expressed as

$$\tilde{G}_{\text{S6}}(\mathbf{u}) = \sum_{i=1}^{d} n^{(i)} \sigma(m^{(i)} F(\mathbf{u}) + \theta_i),$$

where $\mathbf{N} = [n^{(1)} \; \cdots \; n^{(L)}]$. Let $G$ be the function defined in Lemma 1. Then, given any choice of $\mathbf{M}, \mathbf{N}, \boldsymbol{\theta}, \mathbf{A}, \mathbf{B}$, and $\mathbf{C}$, we know that there is a pair of inputs $\mathbf{u}$ and $\mathbf{v}$ such that

$$F(\mathbf{u}) = F(\mathbf{v}), \qquad |G(\mathbf{u}) - G(\mathbf{v})| > 1.$$

Since $F(\mathbf{u}) = F(\mathbf{v})$, we also have that $\tilde{G}_{\text{S6}}(\mathbf{u}) = \tilde{G}_{\text{S6}}(\mathbf{v})$. By the triangle inequality, we have

$$|G(\mathbf{u}) - \tilde{G}_{\text{S6}}(\mathbf{u})| + |G(\mathbf{v}) - \tilde{G}_{\text{S6}}(\mathbf{v})| \geq |G(\mathbf{u}) - G(\mathbf{v})| > 1.$$

Setting $\epsilon = 1/2$, we are done. $\qquad\square$

### B.2 Proof of Theorem 4

The proof of Theorem 4 can be easily established upon our proof that S4D models are universal approximators.

*Proof of Theorem 4.* Note that the construction of a universal approximator in Theorem 1 can be achieved by using the same $\mathbf{C}^{(i)} = [1 \; \cdots \; 1]$ for all $1 \leq i \leq d$ (see [88, 97]). The second statement follows immediately from the first statement of Theorem 1 by setting ${}^1\mathbf{B}_{\text{weight}} = \mathbf{0}$, $\mathbf{B}_{\text{bias}}^{(i)}$ to be the same $\mathbf{B}^{(i)}$ used in $\Gamma_{\text{S4D}}$, and ${}^1\mathbf{C}$ to be the rank-1 matrix whose entries are all $d^{-1}$ so that $u_L \mathbf{M}^\top {}^1\mathbf{C} = [1 \; \cdots \; 1]$. For the first statement, consider the function $H(\mathbf{u})$ given by $H(\mathbf{u}) = G(\sqrt{\mathbf{u}})$, where the square-root is applied elementwise. Clearly, $H$ is continuous since $G$ is continuous. Then, we know, by the second statement, that there exist $\mathbf{M} = [1 \; \cdots \; 1]^\top, \mathbf{N}, \boldsymbol{\theta}, \tilde{\mathbf{A}}, {}^1\tilde{\mathbf{B}}_{\text{weight}} = \mathbf{0}, \tilde{\mathbf{B}}_{\text{bias}}^{(i)}$, and ${}^1\tilde{\mathbf{C}} = [1/d]_{i,j}$ such that the $\text{B}_2\text{S}_6$ system $\tilde{\boldsymbol{\Gamma}}_{\text{B}_2\text{S}_6}$ defined in eq. (7) with $h = 1$ and $p = d$ and the map

$$\tilde{H}_{\text{B}_2\text{S}_6} : [0,1]^L \to \mathbb{R}, \quad (u_1, \ldots, u_L) \mapsto \mathbf{N} \sigma(\tilde{\boldsymbol{\Gamma}}_{\text{B}_2\text{S}_6}((\mathbf{M}u_1, \ldots, \mathbf{M}u_L))_L + \boldsymbol{\theta})$$

satisfy that

$$|\tilde{H}_{\text{B}_2\text{S}_6}(\mathbf{u}) - H(\mathbf{u})| \leq \epsilon, \qquad \text{for any } \mathbf{u} \in [0,1]^{L-1} \times \{1\}.$$

Now, define $h = d$ and $p = 1$. Let $\mathbf{M}, \mathbf{N}, \boldsymbol{\theta}, \mathbf{A} = \tilde{\mathbf{A}}$ be the same matrices, and let ${}^j\mathbf{B}_{\text{weight}} = \tilde{\mathbf{B}}_{\text{bias}}^{(j)}$, $\mathbf{B}_{\text{bias}}^{(i)} = \mathbf{0}$, and ${}^j\mathbf{C} = [1 \; \cdots \; 1]$ for all $i$ and $j$. Then, it is clear that the system $\boldsymbol{\Gamma}_{\text{B}_2\text{S}_6}$ defined by these matrices satisfies that

$$(\boldsymbol{\Gamma}_{\text{B}_2\text{S}_6}(\mathbf{u}))_L = (\tilde{\boldsymbol{\Gamma}}_{\text{B}_2\text{S}_6}(\mathbf{u}^2))_L, \qquad \text{for any } \mathbf{u} \in [0,1]^{L-1} \times \{1\},$$

where $\mathbf{u}^2$ is the sequence obtained by squaring every entry of $\mathbf{u}$. Hence, the map

$$\tilde{G}_{\text{B}_2\text{S}_6} : [0,1]^{L-1} \times \{1\} \to \mathbb{R}, \quad (u_1, \ldots, u_L) \mapsto \mathbf{N} \sigma(\boldsymbol{\Gamma}_{\text{B}_2\text{S}_6}((\mathbf{M}u_1, \ldots, \mathbf{M}u_L))_L + \boldsymbol{\theta})$$

satisfies that

$$|\tilde{G}_{\text{B}_2\text{S}_6}(\mathbf{u}) - G(\mathbf{u})| = |\tilde{H}_{\text{B}_2\text{S}_6}(\mathbf{u}^2) - H(\mathbf{u}^2)| \leq \epsilon, \qquad \text{for any } \mathbf{u} \in [0,1]^{L-1} \times \{1\}.$$

The proof is complete. $\qquad\square$

## C  Proofs of inductive bias results

In this section, we prove Theorem 2 and 5. The proof of these results relies on a very simple argument that helps us avoid the cancellation of terms, which we state below.

**Lemma 2.** Let $p \in \mathbb{R}$ be given. Assume $f : \mathbb{R} \to \mathbb{R}$ and $g : \mathbb{R} \to \mathbb{R}$ are two functions such that $f(x) = \alpha x^p + o(x^p)$ and $g(x) = \beta x^p + o(x^p)$ as $x \to \infty$, for some constants $\alpha, \beta \neq 0$. Then, we have that for a.e. choice of $c_1, c_2 \in \mathbb{R}$ that $c_1 f(x) + c_2 g(x) = \gamma x^p + o(x^p) = \Theta(x^p)$ as $x \to \infty$ for some $\gamma \neq 0$.

*Proof.* The proof is straightforward: as long as we have that $\alpha c_1 + \beta c_2 \neq 0$, the conclusion is satisfied. Since $\alpha$ and $\beta$ are nonzero, the equation $\alpha c_1 + \beta c_2 = 0$ is satisfied only on a Lebesgue null set of $c_1$ and $c_2$. $\qquad\square$

### C.1 Proof of Theorem 2

We now prove the main theorems by explicitly calculating the Jacobians.

*Proof of Theorem 2.* Given an S6 system in eq. (3) and an input sequence $\mathbf{u} = (\mathbf{u}_1, \ldots, \mathbf{u}_L)$, let $\mathbf{y}_L = \Gamma_{S6}(\mathbf{u})$ be the last output of the system. For any $1 \leq s \leq d$, we have that

$$y_L^{(s)} = \mathbf{u}_L^\top \mathbf{C} \mathbf{x}_L = \mathbf{u}_L^\top \mathbf{C} \sum_{j=1}^{L} \left( \prod_{i=j+1}^{L} \mathbf{A}(\mathbf{u}_i) \right) \mathbf{B}(\mathbf{u}_j) u_j^{(s)},$$

where

$$\mathbf{A}(\mathbf{u}_i) = \exp\left(\Delta(\mathbf{u}_i)\mathbf{A}\right) = \exp\left(\text{softplus}(\mathbf{w}^\top \mathbf{u}_i)\mathbf{A}\right),$$
$$\mathbf{B}(\mathbf{u}_i) = \mathbf{A}^{-1}\left(\exp\left(\text{softplus}(\mathbf{w}^\top \mathbf{u}_i)\mathbf{A}\right) - \mathbf{I}\right)\mathbf{B}\mathbf{u}_i.$$

Note that we have changed the notations so that $\mathbf{A}(\mathbf{u}_i)$ is exactly the matrix $\overline{\mathbf{A}}_i$ and $\mathbf{B}(\mathbf{u}_i)$ is exactly the matrix $\overline{\mathbf{B}}_i$ in eq. (3). This helps us better keep track of the dependency of every term on the input $\mathbf{u}$. In addition, it is easy to see that $b^{(i)}$ does not play a role in the asymptotic behaviors when $c \to \pm\infty$. Hence, we set them to zero. For any $s$, we have

$$(\mathbf{J}_r)_{s,s} = \frac{\partial y_L^{(s)}}{\partial u_r^{(s)}} = \mathbf{u}_L^\top \mathbf{C} \sum_{j=1}^{L} \underbrace{\frac{\partial}{\partial u_r^{(s)}}\left[\left(\prod_{i=j+1}^{L} \mathbf{A}(\mathbf{u}_i)\right) \mathbf{B}(\mathbf{u}_j) u_j^{(s)}\right]}_{\mathbf{d}_j^{(r,s)}},$$

where

$$\mathbf{d}_j^{(r,s)} = \begin{cases} \left[\frac{\partial}{\partial u_r^{(s)}} \mathbf{A}(\mathbf{u}_r)\right]\left(\prod_{i=j+1, i\neq r}^{L} \mathbf{A}(\mathbf{u}_i)\right) \mathbf{B}(\mathbf{u}_j) u_j^{(s)} & , j < r, \\ \left(\prod_{i=r+1}^{L} \mathbf{A}(\mathbf{u}_i)\right)\left(\mathbf{B}(\mathbf{u}_r) + u_r^{(s)} \frac{\partial}{\partial u_r^{(s)}} \mathbf{B}(\mathbf{u}_r)\right) & , j = r, \\ \mathbf{0} & , j > r. \end{cases}$$

For any $s \neq t$, we have

$$(\mathbf{J}_r)_{s,t} = \frac{\partial y_L^{(t)}}{\partial u_r^{(s)}} = \mathbf{u}_L \mathbf{C} \sum_{j=1}^{L} \underbrace{\frac{\partial}{\partial u_r^{(s)}}\left[\left(\prod_{i=j+1}^{L} \mathbf{A}(\mathbf{u}_i)\right) \mathbf{B}(\mathbf{u}_j) u_j^{(t)}\right]}_{\mathbf{f}_j^{(r,s,t)}},$$

where

$$\mathbf{f}_j^{(r,s,t)} = \begin{cases} \left[\frac{\partial}{\partial u_r^{(s)}} \mathbf{A}(\mathbf{u}_r)\right]\left(\prod_{i=j+1, i\neq r}^{L} \mathbf{A}(\mathbf{u}_i)\right) \mathbf{B}(\mathbf{u}_j) u_j^{(t)} & , j < r, \\ \left(\prod_{i=r+1}^{L} \mathbf{A}(\mathbf{u}_i)\right) u_r^{(t)} \frac{\partial}{\partial u_r^{(s)}} \mathbf{B}(\mathbf{u}_r) & , j = r, \\ \mathbf{0} & , j > r. \end{cases}$$

We now break the proof into two cases.

**Case I: $c \to \infty$.** We discuss the cases when $r < k$, $r = k$, and $r > k$ separately.

- When $r < k$, it is easy to check that $\|\mathbf{A}(\mathbf{u}_r)\|$ decays exponentially when $c \to \infty$. Hence, we have $\|\mathbf{d}_j^{(r,s)}\|$ and $\|\mathbf{f}_j^{(r,s,t)}\|$ decay exponentially as $c \to \infty$ for every $s$, $t$, and $j$. That is, the Jacobian norm $\|\mathbf{J}_r\|_F$ decays exponentially as $c \to \infty$.

- When $r = k$, the norm of the term

$$\left(\prod_{i=r+1}^{L} \mathbf{A}(\mathbf{u}_i)\right) u_r^{(t)} \frac{\partial}{\partial u_r^{(s)}} \mathbf{B}(\mathbf{u}_r)$$

  grows like $\alpha c + o(c)$ as $c \to \infty$ for all $\mathbf{A}$ and a.e. choice of and $\mathbf{B}$, where $\alpha \neq 0$ is a constant. That is, $\mathbf{d}_k^{(k,s)}$ and $\mathbf{f}_k^{(k,s,t)}$ grow like a linear factor plus a $o(c)$ term as $c \to \infty$. Hence, by Lemma 2, for a.e. choice of $\mathbf{C}$, we have that $\|\mathbf{J}_k\|_F = \Theta(c)$ as $c \to \infty$.

- When $r > k$, it is straightforward to check that $\|\mathbf{d}_k^{(r,s)}\| = \alpha(s)c^2 + o(c^2)$ for some constants $\alpha(s) \neq 0$ and a.e. choice of $\mathbf{B}$ and $\|\mathbf{f}_k^{(r,s,t)}\| = \beta(s,t)c^2 + o(c^2)$ for some constants $\beta(s,t) \neq 0$ and a.e. choice of $\mathbf{B}$. Hence, by Lemma 2, for a.e. choice of $\mathbf{C}$, we have that $\|\mathbf{J}_r\|_F = \Theta(c^2)$ as $c \to \infty$.

The first part of the theorem is proved by combining the three statements above.

**Case II:** $c \to -\infty$. We discuss the cases when $r < k$, $r = k$, and $r > k$ separately.

- When $r < k$, the only thing that changes in the expression of $\mathbf{J}_r$ when $c \to -\infty$ is the matrix $\mathbf{A}(\mathbf{u}_k)$, in which case it converges to $\mathbf{I}$. Hence, we have that $\|\mathbf{J}_r\|_F = \Theta(1)$.

- When $r = k$, we have that

$$\left\| \frac{\partial}{\partial u_k^{(s)}} \mathbf{A}(u_k) \right\| = \mathcal{O}(|c|^{-p})$$

for any $p > 0$ and $s$. Moreover, we also have that

$$\|\mathbf{B}(\mathbf{u}_k)\| = \mathcal{O}(|c|^{-p}), \qquad \left\| \frac{\partial}{\partial u_k^{(s)}} \mathbf{B}(u_k) \right\| = \mathcal{O}(|c|^{-p}),$$

for any $p > 0$ and $s$. That means we have $\|\mathbf{d}_j^{(k,s)}\|$ and $\|\mathbf{f}_j^{(k,s,t)}\|$ decay exponentially for any choice of $s, t$, and $j$. Hence, we have that $\|\mathbf{J}_k\|_F = \Theta(|c|^{-p})$ for any $p > 0$.

- When $r > k$, we note that while $\|\mathbf{u}_k\|$ grows linearly, the norm of the vector $\|\mathbf{B}(\mathbf{u}_k)\|$ decays exponentially. Therefore, we have that $\|\mathbf{B}(\mathbf{u}_k)u_k^{(s)}\|$ decays exponentially for every $s$. This shows that for a.e. $\mathbf{B}$, the norms $\|\mathbf{d}_j^{(r,s)}\|$ and $\|\mathbf{f}_j^{(r,s,t)}\|$ converge to constants for all $s, t$, and $j$. By Lemma 2, we have $\|\mathbf{J}_r\|_F = \Theta(1)$ for a.e. choice of $\mathbf{C}$.

The second part of the theorem is proved by combining the three statements above. The claim about an $\Gamma_{\text{S4D}}$ system is obvious since the system is linear. $\qquad\square$

Interestingly, from the proof of Theorem 2, we see that when $c \to \infty$, the reason why $S_{\text{S6},k}$ is large for $k > k_0$ is that $\mathbf{u}_k$ plays an important role in determining the matrix $\overline{\mathbf{B}}_{k_0}$ that affects a large input $\mathbf{u}_{k_0}$. This is analogous to the function $f(x,y) = xy$. When $x$ is large and $y$ is small, the gradient $(\partial/\partial x)f$ is still small but $(\partial/\partial y)f$ is huge.

## C.2 Proof of Theorem 5

Once Theorem 2 is proved, the proof of Theorem 5 can be maintained easily from it.

*Proof of Theorem 5.* For each $1 \leq r \leq L$, the Jacobian $\mathbf{J}_r$ is a block diagonal matrix $\mathbf{J}_r = \text{diag}(^1\mathbf{J}_r, \ldots, {}^h\mathbf{J}_r)$, where every ${}^j\mathbf{J}_r$ is a $p \times p$ matrix. Since the cases when $c$ is negative and when $c$ is positive are symmetric, without loss of generality, we assume that $c \to \infty$. We study the Jacobian norms when $r < k$, $r = k$, and $r > k$ separately. The following statements hold for a.e. model parameters.

- When $r < k$, from the proof of Theorem 2, we know that $\|^j\mathbf{J}_r\|$ decays exponentially when ${}^j\mathbf{w}^\top {}^j\mathbf{u} > 0$ and $\|^j\mathbf{J}_r\| = \Theta(1)$ when ${}^j\mathbf{w}^\top {}^j\mathbf{u} < 0$. Hence, by our assumption, we know that $\|\mathbf{J}_r\|_F = \Theta(1)$.

- When $r = k$, from the proof of Theorem 2, we know that $\|^j\mathbf{J}_r\| = \Theta(c)$ as long as ${}^j\mathbf{w}^\top {}^j\mathbf{u} > 0$. Hence, we have $\|\mathbf{J}_r\|_F = \Theta(c)$.

- When $r > k$, from the proof of Theorem 2, we know that $\|^j\mathbf{J}_r\| = \Theta(c^2)$ as long as ${}^j\mathbf{w}^\top {}^j\mathbf{u} > 0$. Hence, we have $\|\mathbf{J}_r\|_F = \Theta(c^2)$.

Combining the three cases, we proved the result. $\qquad\square$

# D  Proof of the stability result

We now bring in the last piece of technical details by proving the stability result in Theorem 3. The proof again relies on a tedious expansion of the gradients, which then gives us expressions that are intellectually interesting to analyze with basic probability theory.

*Proof of Theorem 3.* Since we assumed that $d = 1$, we drop all superscripts for channel indices. Given an input $\mathbf{u} = (u_1, \ldots, u_L)$ be an input and denote by $y_{\text{S4D}} = \Gamma_{\text{S4D}}(\mathbf{u})_L$ and $y_{\text{S6}} = \Gamma_{\text{S6}}(\mathbf{u})_L$ the outputs of an S4D system and an S6 system, respectively. Then, we have

$$\frac{\partial y_{\text{S4D}}}{\partial b} = \mathbf{C} \sum_{j=1}^{L} \underbrace{\frac{\partial}{\partial b} \left[ \left( \prod_{i=j+1}^{L} \overline{\mathbf{A}} \right) \overline{\mathbf{B}} u_j \right]}_{\mathbf{r}_j},$$

where

$$\overline{\mathbf{A}} = \exp(\exp(b)\mathbf{A}), \qquad \overline{\mathbf{B}} = \mathbf{A}^{-1}(\overline{\mathbf{A}} - \mathbf{I})\mathbf{B},$$

and

$$\frac{\partial y_{\text{S6}}}{\partial w} = u_L \mathbf{C} \sum_{j=1}^{L} \underbrace{\frac{\partial}{\partial w} \left[ \left( \prod_{i=j+1}^{L} \mathbf{A}(u_i) \right) \mathbf{B}(u_j) u_j \right]}_{\mathbf{s}_j},$$

$$\frac{\partial y_{\text{S6}}}{\partial b} = u_L \mathbf{C} \sum_{j=1}^{L} \underbrace{\frac{\partial}{\partial b} \left[ \left( \prod_{i=j+1}^{L} \mathbf{A}(u_i) \right) \mathbf{B}(u_j) u_j \right]}_{\mathbf{t}_j},$$

where

$$\mathbf{A}(u_j) = \tilde{\mathbf{A}} = \exp(\text{softplus}(b)\mathbf{A}), \qquad \mathbf{B}(u_j) = \mathbf{A}^{-1}(\mathbf{A}(u_j) - \mathbf{I})\mathbf{B}u_j.$$

Note that the matrix $\mathbf{A}(u_j)$ does not really depend on $u_j$ because we assumed that $w = 0$, but we keep the notation consistent with the proof of Theorem 2. In particular, we have that

$$\frac{\partial}{\partial b}\overline{\mathbf{A}} = \overline{\mathbf{A}}\mathbf{A}\exp(b), \qquad \frac{\partial}{\partial b}\overline{\mathbf{B}} = \mathbf{A}^{-1}\overline{\mathbf{A}}\mathbf{A}\exp(b)\mathbf{B} = \overline{\mathbf{A}}\mathbf{B}\exp(b).$$

and

$$\frac{\partial}{\partial w}\mathbf{A}(u) = \frac{\exp\left(\text{softplus}(b)\mathbf{A}\right)\mathbf{A}u}{1 + e^{-b}}, \qquad \frac{\partial}{\partial w}\mathbf{B}(u) = \frac{\exp\left(\text{softplus}(b)\mathbf{A}\right)\mathbf{A}u}{1 + e^{-b}}\mathbf{A}^{-1}\mathbf{B}u,$$

$$\frac{\partial}{\partial b}\mathbf{A}(u) = \frac{\exp\left(\text{softplus}(b)\mathbf{A}\right)\mathbf{A}}{1 + e^{-b}}, \qquad \frac{\partial}{\partial b}\mathbf{B}(u) = \frac{\exp\left(\text{softplus}(b)\mathbf{A}\right)\mathbf{A}}{1 + e^{-b}}\mathbf{A}^{-1}\mathbf{B}u.$$

Using the product rule, we can compute $\mathbf{r}_j$, $\mathbf{s}_j$, and $\mathbf{t}_j$:

$$\mathbf{r}_j = u_j \left( \overline{\mathbf{A}}^{L-j} \frac{\partial}{\partial b}\overline{\mathbf{B}} + \sum_{i=j+1}^{L} \left( \frac{\partial}{\partial b}\overline{\mathbf{A}} \right) \overline{\mathbf{A}}^{L-j-1}\overline{\mathbf{B}} \right)$$

$$= \overline{\mathbf{A}}^{L-j-1} u_j \left( \overline{\mathbf{A}} \frac{\partial}{\partial b}\overline{\mathbf{B}} + \left( \sum_{i=j+1}^{L} \frac{\partial}{\partial b}\overline{\mathbf{A}} \right) \overline{\mathbf{B}} \right)$$

$$= \overline{\mathbf{A}}^{L-j} u_j \exp(b) \left( \overline{\mathbf{A}} + (L-j)(\overline{\mathbf{A}} - \mathbf{I}) \right) \mathbf{B},$$

and

$$\mathbf{s}_j = u_j \left( \tilde{\mathbf{A}}^{L-j} \frac{\partial}{\partial w} \mathbf{B}(u_j) + \sum_{i=j+1}^{L} \frac{\partial}{\partial w} \mathbf{A}(u_i) \tilde{\mathbf{A}}^{L-j-1} \mathbf{B}(u_j) \right)$$

$$= \tilde{\mathbf{A}}^{L-j-1} \left( u_j \, \tilde{\mathbf{A}} \, \frac{\partial}{\partial w} \mathbf{B}(u_j) + u_j \left( \sum_{i=j+1}^{L} \frac{\partial}{\partial w} \mathbf{A}(u_i) \right) \mathbf{B}(u_j) \right)$$

$$= \tilde{\mathbf{A}}^{L-j-1} \frac{\exp{(\mathrm{softplus}(b)\mathbf{A})}}{1+e^{-b}} \left( \tilde{\mathbf{A}} \, \mathbf{B} u_j^3 + u_j^2 \left( \exp{(\mathrm{softplus}(b)\mathbf{A})} - \mathbf{I} \right) \mathbf{B} \sum_{i=j+1}^{L} u_i \right),$$

$$\mathbf{t}_j = u_j \left( \tilde{\mathbf{A}}^{L-j} \frac{\partial}{\partial b} \mathbf{B}(u_j) + \sum_{i=j+1}^{L} \frac{\partial}{\partial b} \mathbf{A}(u_i) \tilde{\mathbf{A}}^{L-j-1} \mathbf{B}(u_j) \right)$$

$$= \tilde{\mathbf{A}}^{L-j-1} \left( u_j \, \tilde{\mathbf{A}} \, \frac{\partial}{\partial b} \mathbf{B}(u_j) + u_j \left( \sum_{i=j+1}^{L} \frac{\partial}{\partial b} \mathbf{A}(u_i) \right) \mathbf{B}(u_j) \right)$$

$$= \tilde{\mathbf{A}}^{L-j-1} \frac{\exp{(\mathrm{softplus}(b)\mathbf{A})}}{1+e^{-b}} \left( \tilde{\mathbf{A}} \, \mathbf{B} u_j^2 + u_j^2 (L-j) \left( \exp{(\mathrm{softplus}(b)\mathbf{A})} - \mathbf{I} \right) \mathbf{B} \right).$$

**Increasing input magnitudes.** From the formulas of $\mathbf{s}_j$, $\mathbf{t}_j$, and $\mathbf{r}_j$, it is straightforward that they are homogeneous in $\mathbf{u}$ with degrees 3, 2, and 1, respectively. Fixing an $L$, as long as $\mathbb{E}_{\mathbf{u}\sim\mathbb{D}_L}|(\partial/\partial w)y_{\mathrm{S6}}|$, $\mathbb{E}_{\mathbf{u}\sim\mathbb{D}_L}|(\partial/\partial b)y_{\mathrm{S6}}|$, and $\mathbb{E}_{\mathbf{u}\sim\mathbb{D}_L}|(\partial/\partial b)y_{\mathrm{S4D}}| \neq 0$, we have that

$$\frac{\mathbb{E}_{\mathbf{u}\sim c\mathbb{D}_L}|(\partial/\partial w)y_{\mathrm{S6}}|}{\mathbb{E}_{\mathbf{u}\sim c\mathbb{D}_L}|(\partial/\partial b)y_{\mathrm{S4D}}|} = \mathcal{O}(c^3), \qquad \frac{\mathbb{E}_{\mathbf{u}\sim c\mathbb{D}_L}|(\partial/\partial b)y_{\mathrm{S6}}|}{\mathbb{E}_{\mathbf{u}\sim c\mathbb{D}_L}|(\partial/\partial b)y_{\mathrm{S4D}}|} = \mathcal{O}(c^2), \qquad c \to \infty.$$

This proves the first part of the theorem.

**Increasing sequence length.** The proof of the theorem when $L \to \infty$ is more involved. For clarity, we break it into three parts.

- **The gradient of $y_{\mathrm{S4D}}$.** We first consider the case when $n = 1$. We will show that $\mathbb{E}_{\mathbf{u}\sim\mathbb{D}_L}|(\partial/\partial b)y_{\mathrm{S4D}}| = \exp(b(L))\mathcal{O}(\sqrt{L})$ as $L \to \infty$. Since $\mathbf{A}$ is diagonal, when $n \geq 1$, $y_{\mathrm{S4D}}$ can be calculated by summing up the outputs from $n$ independent LTI systems, all with a 1-dimensional state space. This obviously shows that $\mathbb{E}_{\mathbf{u}\sim\mathbb{D}_L}|(\partial/\partial b)y_{\mathrm{S4D}}| = \exp(b(L))\mathcal{O}(\sqrt{L})$ when $n \geq 1$. To prove the case when $n = 1$, we use Jensen's inequality and obtain that

$$\mathbb{E}\left[ \left| \sum_{j=1}^{L} \overline{\mathbf{A}}^{L-j} \mathbf{B} u_j \right| \right] \leq \sqrt{ \mathbb{E}\left[ \left| \sum_{j=1}^{L} \overline{\mathbf{A}}^{L-j} \mathbf{B} u_j \right|^2 \right] }$$

$$= \sqrt{ \mathbb{E}\left[ \sum_{j=1}^{L} \overline{\mathbf{A}}^{L-j} \mathbf{B} u_j \right]^2 + \mathrm{Var}\left[ \sum_{j=1}^{L} \overline{\mathbf{A}}^{L-j} \mathbf{B} u_j \right] }$$

$$= \sqrt{ 0 + \sum_{j=1}^{L} \mathrm{Var}(v_j) + \sum_{1 \leq i < j \leq L} \mathrm{Cov}(v_i, v_j) } = \mathcal{O}(\sqrt{L}),$$

where $v_j = \overline{\mathbf{A}}^{L-j} \mathbf{B} u_j$. This shows that

$$\mathbb{E}_{\mathbf{u}\sim\mathbb{D}_L}\left[ \left| \frac{\partial}{\partial b} y_{\mathrm{S4D}} \right| \right] = \mathbb{E}_{\mathbf{u}\sim\mathbb{D}_L}\left[ \left| \mathbf{C} \sum_{j=1}^{L} \mathbf{r}_j \right| \right]$$

$$\leq \exp(b(L)) \left( \mathbb{E}_{\mathbf{u}\sim\mathbb{D}_L}\left[ \left| \mathbf{C} \sum_{j=1}^{L} \overline{\mathbf{A}}^{L-j+1} \mathbf{B} u_j \right| \right] + \mathbb{E}_{\mathbf{u}\sim\mathbb{D}_L}\left[ \left| \mathbf{C} \sum_{j=1}^{L} \overline{\mathbf{A}}^{L-j} (L-j)(\overline{\mathbf{A}} - \mathbf{I}) \mathbf{B} u_j \right| \right] \right).$$

Note that we just proved that the first expectation is $\mathcal{O}(\sqrt{L})$ as $L \to \infty$. By letting $b(L) \to -\infty$ fast enough, $\overline{\mathbf{A}} - \mathbf{I}$ vanishes exponentially, so we have the second term is also $\mathcal{O}(\sqrt{L})$. This proves that

$$\mathbb{E}_{\mathbf{u} \sim \mathbb{D}_L} |(\partial/\partial b) y_{\text{S4D}}| = \exp(b(L)) \mathcal{O}(\sqrt{L}), \qquad L \to \infty. \tag{10}$$

- **The gradient of $y_{\text{S6}}$ when $n = 1$.** When $n = 1$, the matrices $\tilde{\mathbf{A}}, \mathbf{B}$, and $\mathbf{C}$ are all scalars. Without loss of generality, assume that $\mathbf{B}, \mathbf{C} > 0$. Similar to the previous case, if $b(L) \to -\infty$ as $L \to \infty$, the matrix $\exp(\text{softplus}(b(L))\mathbf{A}) - \mathbf{I}$ vanishes. That is, since $u_L = 1$, we have

$$\mathbb{E}_{\mathbf{u} \sim \mathbb{D}_L} \left[ \left| \frac{\partial}{\partial b} y_{\text{S6}} \right| \right] = \mathbb{E}_{\mathbf{u} \sim \mathbb{D}_L} \left[ \left| \mathbf{C} \sum_{j=1}^{L} \mathbf{t}_j \right| \right] \sim \exp(b(L)) \mathbb{E}_{\mathbf{u} \sim \mathbb{D}_L} \left[ \left| \mathbf{C} \sum_{j=1}^{L} \tilde{\mathbf{A}}^{L-j} \mathbf{B} u_j^2 \right| \right]$$

$$= \exp(b(L)) \sum_{j=1}^{L} \mathbf{C} \tilde{\mathbf{A}}^{L-j} \mathbf{B} \, \mathbb{E} \left[ u_j^2 \right] \geq \exp(b(L)) \sum_{j=1}^{L} \mathbf{C} \tilde{\mathbf{A}}^{L-j} \mathbf{B} \, \text{Var} \left[ u_j \right] = \exp(b(L)) \Theta(L),$$

provided that $b(L)$ decays fast enough so that $\tilde{\mathbf{A}}^L = \Theta(1)$ as $L \to \infty$.

- **The gradient of $y_{\text{S6}}$ when $n > 1$.** Now, suppose $n > 1$. The output $y_{\text{S6}}$ can be written as the sum of $n$ terms: $y_{\text{S6}} = y_1 + \cdots + y_n$, where $y_j$ is the output of an LTI system with $n = 1$ [97]. By the previous argument, we know that for every $1 \leq j \leq n$,

$$\mathbb{E}_{\mathbf{u} \sim \mathbb{D}_L} \left[ \frac{\partial}{\partial b} \frac{y_j}{\mathbf{C}_j} \right] = \pm \exp(b(L)) \Theta(L),$$

where the expectation can be either positive or negative, depending on the sign of $\mathbf{B}_j \mathbf{C}_j$. That is, we know that

$$\mathbb{E}_{\mathbf{u} \sim \mathbb{D}_L} \left[ \left| \frac{\partial}{\partial b} \frac{y_j}{\mathbf{C}_j} \right| \right] \Big/ (\exp(b(L)) L)$$

is bounded between two positive numbers or between two negative numbers as $L$ is sufficiently large. We still want to apply Lemma 2 to control cancellation effects, but the main issue is that we do not have that this term converges to a number. However, since the metric space $\mathbb{R}^n$ is complete, i.e., boundedness implies subsequent convergence, we know that there exists a subsequence $L_1, L_2, \ldots$ such that

$$\mathbb{E}_{\mathbf{u} \sim \mathbb{D}_{L_k}} \left[ \frac{\partial}{\partial b} \frac{y_j}{\mathbf{C}_j} \right] \to \alpha_j \exp(b(L_k)) L_k \qquad \text{as} \qquad k \to \infty, \qquad 1 \leq j \leq n,$$

where $\alpha_j$ is a nonzero constant for all $1 \leq j \leq n$. Hence, by Lemma 2, we have for a.e. $\mathbf{C}$ that

$$\mathbb{E}_{\mathbf{u} \sim \mathbb{D}_{L_k}} \left[ \frac{\partial y_{\text{S6}}}{\partial b} \right] = \mathbb{E}_{\mathbf{u} \sim \mathbb{D}_{L_k}} \left[ \sum_{j=1}^{n} \frac{\partial y_j}{\partial b} \right] = \alpha \exp(b(L_k)) L_k, \qquad \alpha \neq 0.$$

Hence, we have

$$\mathbb{E}_{\mathbf{u} \sim \mathbb{D}_{L_k}} \left[ \left| \frac{\partial y_{\text{S6}}}{\partial b} \right| \right] = \alpha \exp(b(L_k)) \Theta(L_k). \tag{11}$$

Combining eq. (10) and (11), we prove the theorem. $\qquad \square$

## E  Supplementary experiments on the stability

In this section, we present two experiments to corroborate our discussion of training stability in section 5. The first experiment numerically verifies Theorem 3 as the input magnitude and the sequence length grow. The second experiment empirically shows that a reduced learning rate on $\Delta$-related parameters help stabilize the training of an S6 model.

### E.1 A numerical experiment that verifies Theorem 3

In this experiment, we set $d = 1$ and randomly sample diag($\mathbf{A}$) from the left half of the complex plane, and randomly sample $\mathbf{B}$ and $\mathbf{C}$. We then use these matrices to construct an S4D system and an S6 system. We sample our length-$L$ input sequences from i.i.d. Gaussian distributions with mean zero and standard deviation $c$.[5] We fix the ratio $L / \exp(b(L))^{-1}$ to take into account the fact that as $L$ increases, we need a longer memory window to capture the long-range dependency. We then compute the quantities in Theorem 3. For the first two experiments, we fix $L = 100$ and let $c$ increase. Then, we fix $c = 1$ and let $L$ increase. The results are shown in Figure 6. All results are averaged over 30 different trials.

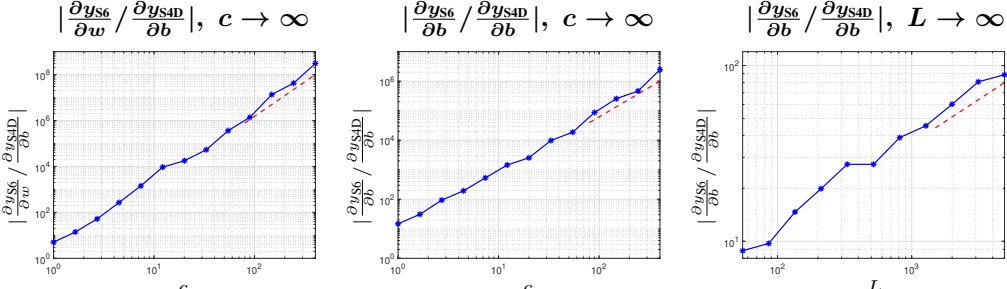

Figure 6: Numerical experiments to verify Theorem 3, where we compute the ratio between the gradients with respect to the S6 parameters and S4D parameters. For the first two figures, we fix $L = 100$; for the last figure, we fix $c = 1$. The gradients are computed using closed algebraic formulas. The red reference lines in the three log-log plots have slopes of 3, 2, and $1/2$, respectively.

In these experiments, we see that the three ratios studied in Theorem 3 follow exactly the pattern of a cubic, quadratic, and square root growth, respectively. This is what we expect from Theorem 3.

### E.2 An empirical experiment on the training stability

We also show a real training example, where we want to learn a ground-truth function that takes in a univariate sequential input and returns a fixed linear combination of them: $G(u_1, \ldots, u_L) = \sum_{j=1}^{L} \theta_j u_j$, where $\theta_1, \ldots, \theta_L$ are fixed parameters. We show the loss curves in Figure 7, with different models (i.e., S4D versus S6) and different learning rate assignments (i.e., whether or not $\Delta$ is learned). We see that the S4D model is very stable during training, with its loss going down smoothly. This is not the case for the S6 model, where the loss goes up and down and even restarts. We find that by freezing the $\Delta t$ parameters $\mathbf{w}$ and $b$, the model trains much more robustly.

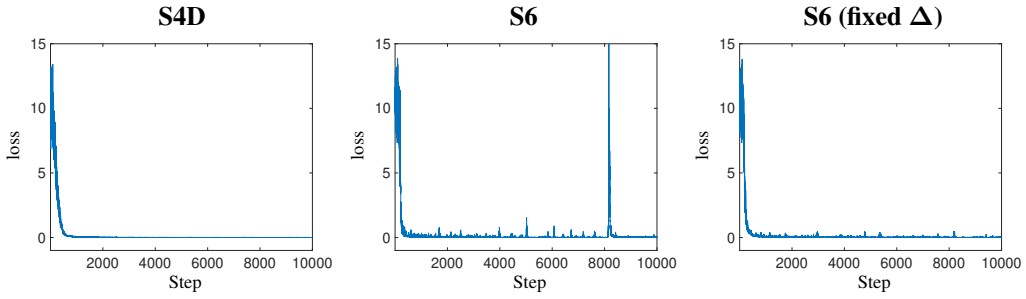

Figure 7: The root-mean-squared loss $\|G(\mathbf{u}) - \tilde{G}(\mathbf{u})\|_2$ between the true output $G(\mathbf{u})$ and the model prediction $\tilde{G}(\mathbf{u})$. The first two models are trained with a learning rate of $0.001$ on the $\Delta$ parameters, whereas the last model is trained with no training of the $\Delta$ parameters.

---

[5]Note, in particular, that this sequence has no long-range dependency. We made this choice because it is a natural one without any prior information. It does not affect the training stability a lot. One can try different input functions with long-range dependencies, e.g., we have tried sinusoidal waves, and the results are similar.

## F Details of experiments

In this section, we show the details of the experiments in section 7. Before we provide the detailed configurations, we provide an extended table for LRA results of more variants of state-space models (see Table 4). While many of these models focus on improving certain aspects of the recurrent unit, we do not incorporate and test them in our $B_2S_6$ model.

Table 4: An extended list of SSM accuracies on the Long-Range Arena benchmark. An entry is left blank if no result is found. All but the last three models are not input-selective.

| Model | ListOps | Text | Retrieval | Image | Pathfinder | Path-X | Avg. |
|---|---|---|---|---|---|---|---|
| S4 [28] | 59.60 | 86.82 | 90.90 | 88.65 | 94.20 | 96.35 | 86.09 |
| S4D [27] | 60.47 | 86.18 | 89.46 | 88.19 | 93.06 | 91.95 | 84.89 |
| DSS [29] | 57.60 | 76.60 | 87.60 | 85.80 | 84.10 | 85.00 | 79.45 |
| LRU [55] | 89.00 | 60.20 | 89.40 | 89.90 | 95.10 | 94.20 | 86.30 |
| S4++ [61] | 57.30 | 86.28 | 84.82 | 82.91 | 80.24 | - | - |
| HOPE-SSM [98] | 62.60 | 89.83 | 91.80 | 88.68 | 95.73 | 98.45 | 87.85 |
| S4D-FT [97] | 62.75 | 89.76 | 92.45 | 90.89 | 95.89 | 97.84 | 88.26 |
| Reg. S4D [46] | 61.48 | 88.19 | 91.25 | 88.12 | 94.93 | 95.63 | 86.60 |
| Liquid S4 [30] | 62.75 | 89.02 | 91.20 | 89.50 | 94.80 | 96.66 | 87.32 |
| RTF SSM [56] | 61.59 | 89.72 | 92.04 | 90.51 | 96.11 | 96.32 | 87.71 |
| Spectral SSM [1] | 60.33 | 89.60 | 90.00 | - | 95.60 | 90.10 | - |
| Spiking SSM [72] | 60.23 | 80.41 | 88.77 | 88.21 | 93.51 | 94.82 | 84.33 |
| S5 [77] | 62.15 | 89.31 | 91.40 | 88.00 | 95.33 | 98.58 | 87.46 |
| S6 (Mamba) [25] | 38.02 | 82.98 | 72.14 | 69.82 | 69.26 | 67.32 | 66.59 |
| S7 [79] | 63.77 | 87.22 | 91.80 | 61.14 | 65.62 | 61.50 | 71.82 |
| $B_2S_6$ (ours) | 63.85 | 88.32 | 91.44 | 88.81 | 95.93 | 97.90 | 87.71 |

In this paper, all models are trained with one or more NVIDIA L40 GPUs with 48GB of memory. For the ablation study, we use a 4-layer model and we increase the number of layers for the full experiments on the LRA benchmark. We provide the details of the model and training hyperparameters used for training each LRA task in Table 5. For all experiments, we set $h = 8$ so that $p = $ #Features$/8$. Notably, this hyperparameter $h$ is not carefully fine-tuned but rather picked randomly. Note that our model for training the `Path-X` tasks is smaller than the corresponding S4D model.

Table 5: Configurations of our $B_2S_6$ model on the LRA benchmark, where LR, BS, and WD stand for learning rate, batch size, and weight decay, respectively.

| Task | Depth | #Features | Norm | Prenorm | LR | BS | Epochs | WD |
|---|---|---|---|---|---|---|---|---|
| ListOps | 8 | 128 | BN | False | 0.008 | 32 | 120 | 0.03 |
| Text | 6 | 256 | BN | True | 0.01 | 16 | 40 | 0.05 |
| Retrieval | 6 | 128 | BN | True | 0.004 | 60 | 200 | 0.03 |
| Image | 6 | 512 | LN | False | 0.01 | 48 | 500 | 0.05 |
| Pathfinder | 6 | 256 | BN | True | 0.004 | 48 | 250 | 0.03 |
| Path-X | 6 | 128 | BN | True | 0.001 | 24 | 120 | 0.03 |

## G Limitations and Future Work

We acknowledge (and justify) several limitations of this work, which also suggest promising directions for future research:

- The universal approximation theorems (UATs) in this paper are derived under a few simplifying assumptions — most notably, the assumption that the sampling interval $\Delta$ is fixed. This setup diverges from practical implementations, where $\Delta$ is input-dependent. However, our goal is not to prove UATs for all realistic settings, but to expose fundamental expressiveness gaps between S6 and S4D under clean conditions. Extending these results to settings with dynamic $\Delta$ remains an interesting theoretical direction. If one can show that a single-layer Mamba is not a universal

approximator even when $\Delta$ is not fixed, then that would further stress the benefit of a multi-head design in selective SSMs. Conversely, if a single-layer Mamba with dynamic $\Delta$ is a universal approximator, then an important follow-up question is if (and how) a trainable channel-specific $\Delta$ can compensate for the channel-independent $\mathbf{B}$ and $\mathbf{C}$ matrices in a single-head Mamba.

- Our experiments on language modeling are not designed to compete with large-scale models or datasets. This choice reflects a deliberate focus: our primary aim is to study and improve the behavior of Mamba on long-range sequence tasks. Demonstrating that $B_2S_6$ retains Mamba's performance in language modeling — even at small scale — supports its versatility. Scaling to larger models and corpora is a natural next step, pending access to greater computational resources.

- Our implementation of $B_2S_6$ is based entirely on PyTorch. While this makes the method easily accessible and reproducible, further speed improvements would require low-level optimization in CUDA (see [25]). We leave efficient implementation and integration into high-performance inference/training libraries to future work.

