# OpenReview forum: "Block-Biased Mamba for Long-Range Sequence Processing"
_NeurIPS.cc/2025/Conference — NeurIPS 2025 poster_

### Official Review · Reviewer_PVYx · 2025-06-26

**Clarity:** 3
**Significance:** 3
**Originality:** 3
**Rating:** 5
**Confidence:** 4

**Summary:**

This paper introduces Block-Biased-S6 (B2S6), a modification of the S6 recurrent layer from Mamba, that adds a channel specific bias term to the linear map acting on the input sequence, $\mathbf{B}_w\mathbf{u}_k$$\rightarrow$$\mathbf{B}_w\mathbf{u}_k+\mathbf{B}_b$. Additionally, the model adopts the block unit design suggested in the Mamba2 paper.

The paper motivates these changes through a number of theoretical results and toy experiments, which explore the expressivity, inductive bias, and training stability of Mamba. They also demonstrate that B2S6 achieves superior performance on the long range arena benchmark compared to other selective state-space models.

**Questions:**

## Questions

1) Can a result similar to Theorem 2 be proved when you assume that a single term in the input dominates, whilst the norm of the input remains fixed?

## Suggestions

1) The discussion of the theoretical results should be modified to align better with the contents of the Theorems. Most critically, I do not believe your theoretical results are sufficient to claim that "Mamba does not have universal approximation properties" (line 35) or that "S6 has structurally limited width compared to S4D" even if $\Delta^{(i)}_k$ is dynamic (line 949). Given this adjustment, I would be happy to raise my score.

**Ethical Concerns:**

["NO or VERY MINOR ethics concerns only"]

**Final Justification:**

The authors have resolved my key concern, which was that their claims about Mamba were not sufficiently backed up by their theoretical results. Given the interesting theoretical results, strong empirical results, and quality of the manuscript, I am now happy to recommend acceptance.

**Limitations:**

Yes.

**Quality:**

3

**Strengths And Weaknesses:**

## Strengths

- The paper is well written, I found no errors in the proofs of the theoretical results, and the theoretical results are explored empirically using suitable toy experiments.
- The ablation study is a good demonstration of the benefits of including the bias term and the trade-off between the number of independent blocks and the dimension of the input to each block.
- The empirical results on the long range arena benchmark demonstrate the benefits of using a block unit design and including the bias term $\mathbf{B}_b$.

## Weaknesses

### Theory

The theoretical results of the paper are interesting, but the resulting claims about Mamba are overstated. In particular:
- "A single-layer Mamba is not a universal approximator" does not follow from Theorem 1. In Theorem 1, the channel specific dynamics of Mamba have been removed by fixing the $\Delta^{(i)}_k$, reducing the model to being identical copies in each channel. As the author's note on line 124, this obviously hurts expressivity. Removing the channel specific dynamics from Mamba also weakens Theorem 4 and the motivation for the channel specific bias in B2S6, as you are adding in an alternative type of channel specific dynamics, motivated by removing the channel specific dynamics that were already present. From the empirical results in Figure 2 and Table 2, the channel specific dynamics from $\Delta^{(i)}_k$ and $\mathbf{B}_b$ clearly have different effects, but this is not explored theoretically.
- Similarly, "Mambas exhibit strong inductive bias" does not follow from Theorem 2, which relies on one diverging sample $c\mathbf{u}_{k_0}$, a case which does not arise in practice due to normalisation. Theorem 2 would be much stronger if you could instead assume that one term dominates the others whilst the norm of the input remains fixed.

Additionally, on line 672 you state that "To the best of our knowledge, however, very little has been done to establish universal approximation properties for Mamba models". However, a paper you cite, [52], established the explicit expressivity of Mamba in the continuous setting without restricting the model to fixed $\Delta^{(i)}_k$. There should be a discussion of your expressivity results in relation to the results presented in [52].

### Experiments

- The bold and underlined results in Table 2 are slightly disingenuous, given that the three models which achieve a higher average test accuracy are relegated to the Appendix. However, this does not weaken the claim of the paper, as B2S6 is still the best performing selective state-space model.

---

> ### Author Rebuttal · Authors · 2025-07-30
>
> We thank the reviewer for the careful review and insightful comments. Below we address the questions and comments raised in this review.
>
> * ### Overclaims in Theorem Discussions (W1 + Suggestion)
>
>   We totally agree that the fixed $\Delta$ in Theorem 1 is a strong assumption, and that a few overclaims were made surrounding it. Since we are unable to submit a rebuttal revision, we outline our planned modifications below based on your suggestion. We hope these adjustments resolve your concern.
>
>   1. Line 32-35 will be changed into
>
>   > **Expressiveness.** Unlike S4D, which allows each internal channel to learn an independent recurrent unit, Mamba shares parameters across all channels. Therefore, one should not interpret Mamba as a straightforward extension of S4D that gains time-variant benefits at no cost. Rather, Theorem 1 shows that while Mamba introduces input-dependent dynamics, it also sacrifices certain degrees of freedom. Most notably, it has a weaker ability to learn independent behavior in each channel.
>
>   2. Line 948-950 will be changed into
>
>   > Extending these results to settings with dynamic $\Delta$ remains an interesting theoretical direction. If one can show that a single-layer Mamba is not a universal approximator even when $\Delta$ is not fixed, then that would further stress the benefit of a multi-head design in selective SSMs. Conversely, if a single-layer Mamba with dynamic $\Delta$ is a universal approximator, then an important follow-up question is how a trainable channel-specific $\Delta$ can compensate for the channel-independent $\mathbf{B}$ and $\mathbf{C}$ matrices in a single-head Mamba.
>
>   In addition to the two cases pointed out by the reviewer, we plan for the following two changes as well:
>
>   3. Line 124-125 will be changed into
>
>   > Rather, Theorem 1 highlights that an S6 unit is not a simple extension of S4D that inherits all of its structural principles, and the sharing of the $\mathbf{B}$ and $\mathbf{C}$ matrices across all channels can reduce its capacity, making it potentially less effective than S4D for certain tasks.
>
>   4. Before Line 128, we will add the following sentence:
>
>   > While Theorem 1 assumes that $\Delta$ is fixed, we use a synthetic experiment to demonstrate that even when $\Delta$ is trainable, the channel-independent design of $\mathbf{B}$ and $\mathbf{C}$ in S6 limits its ability to solve certain problems than S4D.
>
>   We also appreciate the reviewer for pointing out [52] as a related work to our expressiveness discussion. We will change Line 672-673 into
>
>   > There are limited works on the universal approximation properties of Mamba models. One notable exception is [52], where the uniform closure of Mamba (i.e., functions that can be uniformly approximated by Mamba) is analyzed in a continuous setting (i.e., the inputs are continuous time series instead of discrete sequences) through linear controlled differential equations. The analysis there highlights the important role of an input-dependent sampling interval $\Delta$ in the expressiveness of Mamba, suggesting that Theorem 1 may not hold for a trainable $\Delta$ (see Appendix H).
>
> * ### Comparison with Selective SSMs (W3)
>
>   We thank the reviewer for pointing out this rendering issue in Table 2. We agree that the bold and underlined numbers do not take other SSMs into account. We will add the following sentence to the caption:
>
>   > While we compare B2S6 only to a few SSM prototypes in this paper, there are many variants of SSMs that can achieve slightly better performance (see Table 3).
>
>   In addition, we have also evaluated Mamba2 on the LRA benchmark during the rebuttal period. We will update the manuscript with the following sub-table:
>
>   | | ListOps | Text | Retrieval | Image | Pathfinder | Path-X | Avg. |
>   | --- | :---: | :---: | :---: | :---: | :---: | :---: | :---: |
>   | Mamba |38.02| 82.98| 72.14| 69.82| 69.26| 67.32| 66.59|
>   | Mamba2 |41.45| 86.09| 79.23| 71.96 | 75.45| 69.07 | 70.54 |
>   | B2S6 |63.85| 88.32| 91.44| 88.81| 95.93| 97.90| 87.71|
>
>   We hope this comparison makes our evaluation more comprehensive.
>
> * ### Variants of Theorem 2 (W2 + Q1)
>
>   This is another great point. You may have noticed that the distinction between a time-variant system in Mamba and a time-invariant system in S4D in the context of relative gradients comes from two sources:
>
>   1. The system in S4D is linear while the system in S6 is quadratic (ignoring the trainable $\Delta$ factor). The former linearity makes the gradient norm independent of the input $\mathbf{u}_k$ while the latter quadratic dependency makes the gradient norm dependent on $\\|\mathbf{u}\_k\\|_2$.
>
>   2. In S6, the input-dependent sampling interval $\Delta(c\mathbf{u}_k)$ increases exponentially when $c$ grows to one end of $\pm \infty$ and decays exponentially when $c$ grows to the other end.
>
>   The reason why Theorem 2 involves an exponential gap is the second source above. That said, if we fix the norm of the entire input sequence, increasing the ratio between $\\|\mathbf{u}\_{k_0}\\|\_2$ and $\\|\mathbf{u}\_{k'}\\|\_2$ by decreasing the magnitude of $\mathbf{u}\_{k'}$, then the sampling interval $\Delta(\mathbf{u}_k)$ will be in a bounded range for all $1 \leq k \leq L$. So asymptotically, we will not see an exponential gap under this setting. The first source above is still relevant under this setting, meaning that as we increase the ratio $c = \\|\mathbf{u}\_{k_0}\\|\_2 / \\|\mathbf{u}\_{k'}\\|\_2$, the ratio between the gradient norms $S\_{\text{S6},k_0} / S\_{\text{S6},k'}$ will be asymptotically proportional to $c$.
>
>   While normalization weakens the theoretical merits of Theorem 2, we highlight an additional limitation in the original S6 block proposed in [25]: in the computation of $\Delta_k^{(i)} = \text{softplus}(\mathbf{w}^\top \mathbf{u}_k + b^{(i)})$, the vector $\mathbf{w}$ defines two half-spaces in $\mathbb{R}^d$. Inputs $\mathbf{u}_k$ that lie in the positive half-space produce larger values of $\Delta$, leading to slower dynamics and better memorization, whereas those in the negative half-space yield smaller $\Delta$, resulting in faster forgetting. This setup restricts the model’s ability to assign distinct memorization biases to different nonlinear or even linear regions of the input space. As discussed in the manuscript, modern implementations of Mamba often replace the vector $\mathbf{w}$ with a low-rank projection, which allows for a finer partitioning of $\mathbb{R}^d$. From this perspective, the multi-head design in Mamba can be seen as another architectural strategy to allocate different memorization biases to different subspaces. We will incorporate a fair and clear discussion of these points into the manuscript.
>
> Thank you again for your careful review and constructive comments. We are committed to improving this manuscript, and please let us know if you have any other questions!

---

> > ### Comment · Reviewer_PVYx · 2025-08-04
> > **Rebuttal Reply**
> >
> > Thank you very much for your thorough rebuttal.
> >
> > Overall, I am happy with the suggested changes and believe they strengthen the paper.
> >
> > My only remaining request is to modify the Section 3 heading. Section headings naturally lack context, and as written, Section 3’s heading serves as the paper’s strongest statement of “A single-layer Mamba is not a universal approximator.” I would prefer a heading that reflects the nuance of the discussion in that section. Given this modification, I will be happy to recommend acceptance.

---

> > > ### Author Response · Authors · 2025-08-04
> > > **Thank you for reviewing our rebuttal and the follow-up comment!**
> > >
> > > Dear reviewer,
> > >
> > > Thank you very much for reading carefully through our rebuttal. We totally agree that the heading of section 3 sounds too strong. We propose to change it to "Expressiveness Trade-offs in Mamba: Temporal Adaptivity versus Channel Independence." We are happy to incorporate any further suggestions you may have; otherwise, we will make all changes outlined in our rebuttal and this response in the camera-ready submission if the paper is accepted.
> > >
> > > Thank you again for your great comments!
> > >
> > > Authors

---

> > > > ### Comment · Reviewer_PVYx · 2025-08-05
> > > >
> > > > Thank you for your prompt response. Given this change, I am happy to recommend this paper for acceptance.

---

> > > > > ### Author Response · Authors · 2025-08-05
> > > > >
> > > > > Thank you once again for your insightful feedback and support in our paper! We will certainly incorporate all changes outlined above into our manuscript.

---

### Official Review · Reviewer_y9vW · 2025-07-02

**Clarity:** 4
**Significance:** 3
**Originality:** 3
**Rating:** 5
**Confidence:** 3

**Summary:**

This paper identifies and addresses a critical limitation of Mamba models, their underperformance on long-range sequence tasks despite architectural suitability. The authors attribute this to three factors: limited expressiveness due to channel-wise parameter sharing (Theorem 1), extreme input-dependent inductive bias causing premature forgetting of long-range information (Theorem 2), and training instability on long sequences (Theorem 3). To resolve these, they propose Block-Biased S6 (B2S6), which partitions inputs into blocks for multi-head selective dynamics and adds channel-specific bias terms. Theoretically, B2S6 achieves universal approximation (Theorem 4) and balanced inductive bias (Theorem 5). Empirically, it outperforms S4/S4D on Long-Range Arena (LRA) (Table 2) while matching Mamba’s language modeling performance on SlimPajama.

**Questions:**

1. Mamba incorporates parallel computing for scaling to larger language model training. Could B2S6's design hinder parallel computation?

2. Could B2S6 match or surpass Mamba in large-scale language modeling (e.g., 1B+ parameters, 100B+ tokens)?

3. Would the universal approximation of B2S6 (Theorem 4) hold with input-dependent Δ (as in standard Mamba)?

4. Section 5 attributes instability to $\Delta$ optimization. Does freezing other parameters also help?

5. Does Mamba2 suffer from the same Expressiveness, Inductive Bias, and Training Stability issues? Could the same toy model experiments (like those in Figures 2 and 3) be used to test Mamba2?

**Ethical Concerns:**

["NO or VERY MINOR ethics concerns only"]

**Final Justification:**

The author addressed my concern. I recommend accepting the paper.

**Limitations:**

Yes

**Paper Formatting Concerns:**

No Formatting Concerns

**Quality:**

4

**Strengths And Weaknesses:**

**Strengths**:
1. The paper provides a comprehensive theoretical diagnosis of Mamba’s limitations, supported by formal theorems on expressiveness, bias, and stability. The non-universal approximation proof for S6 (Theorem 1) and the bias characterization (Theorem 2) are particularly insightful.
2. B2S6 elegantly combines block-wise processing with channel-specific bias, addressing all three limitations. Theorems 4–5 formally validate its expressiveness and bias correction.
3. B2S6 achieves competitive performance on LRA (e.g., +18.5% avg. gain over Mamba in Table 2) while preserving language modeling efficacy (Fig. 7). Ablations (Table 1) and synthetic tasks (Figs. 2–3) robustly justify design choices.
4. Detailed proofs (Appendices B–D), hyperparameters (Table 4), and open code enhance reproducibility.
5. The toy model experiments and theory validate each other very well.

**Weakness:**:
1. Experiments use a small SlimPajama subset (6B tokens) and a 250M-parameter model. Validation on larger model scales is needed to confirm generality.

2. Theorems assume fixed $\Delta$, while practical Mamba uses input-dependent $\Delta$. While simplifying proofs, this may weaken the real-world relevance of expressiveness claims.

3. The computational cost of block partitioning and bias terms is unquantified. Comparisons against efficient S4/S4D and Mamba in latency/throughput would strengthen practicality.

4. minor typos: Lines 157 & 159: Standardized "k'th" and "kth" → "k-th".

---

> ### Author Rebuttal · Authors · 2025-07-30
>
> We thank the reviewer for the careful review and insightful comments. Below we address the questions and comments raised in this review.
>
> * ### Efficiency of B2S6 (Q1 + W3)
>
>   We report the throughputs (tokens/sec) of Mamba and B2S6 models below. The block design in B2S6 does not hinder parallel computing, as shown in the Mamba2 paper [20]. The bias term $\mathbf{B}\_{\text{bias}}$ is benign either, because the Mamba algorithm computes the discrete matrices ${^j}\overline{\mathbf{A}}$ and $^j\overline{\mathbf{B}}$ of shape $(B \\; L \\; p \\; n)$ (see Algorithm 2) and one can simply broadcast $^j\mathbf{B}_{\text{bias}}$ to $^j\mathbf{B}$ to achieve this. In the original Mamba implementation, however, this step is done in the CUDA code; our implementation right now is purely PyTorch and does not involve any CUDA optimization. The throughputs reported below are based on comparing pure PyTorch implementations of the models. We mentioned this in the limitation section.
>
>   | Model Size | Mamba | B2S6 |
>   | :--- | :---: | :---: |
>   | 250M | 14,667 | 14,218 |
>   | 370M | 9,914 | 9,676 |
>   | 500M | 7,218 | 7,010 |
>
> * ### Relaxing the Fixed $\Delta$ Assumption (Q3 + W2)
>
>   It is an interesting question whether Mamba with a trainable $\Delta$ is a universal approximator. While relaxing $\Delta$ breaks our non-UAT proof, it is hard either to directly leverage some conventional wisdom to prove a UAT result for Mamba with trainable $\Delta$, as we did for S4D models. Nonetheless, as we mentioned in the manuscript, whether or not a single-layer Mamba (with or without a fixed $\Delta$) is a universal approximator is not a practically important question anyway; instead, the moral of the Theorem 1 is that it highlights that a single-head Mamba, while extends the S4D model by making the system time-variant, also loses the channel-dependent $\mathbf{B}$ and $\mathbf{C}$; therefore, they should not be thought of as a direct extension of S4Ds. This causes some convergence, generalization, and training stability issues, as pointed out on Line 122-127. For the theoretical community, it is indeed important to stress that the non-UAT result we proved holds only in a restricted setting -- we thank the reviewer for bringing this up and will be sure to include a more justified discussion of the theorem (see our response to Reviewer PVYx).
>
> * ### Stability of B2S6 (Q4)
>
>   This is another very interesting question. In general, freezing some of the other parameters indeed helps stabilize the training, but it also puts us at risk of converging to bad minima. We have done some empirical analysis and would be happy to provide some general observations we have made:
>
>   1. It never works to freeze $\mathbf{B}$ and $\mathbf{C}$. These two matrices are in some sense the most crucial ones that define a system and play the same role as the weight matrix $\mathbf{W}$ in an MLP, and freezing these two parameters will lead to a significant performance drop.
>
>   2. For LRA tasks, freezing the matrix $\mathbf{A}$ usually does not impact the performance by a lot. For instance, if the matrix $\mathbf{A}$ is properly initialized using HiPPO-LegS [26] or its scaled variants [96], then the performance is dropped by less than $2\\%$ when $\mathbf{A}$ is frozen. In the meantime, as pointed out in [26], freezing $\mathbf{A}$ or reducing its learning rate indeed stabilizes the training. For pretraining LLMs, training the matrix $\mathbf{A}$ rarely causes a stability issue, so there is no need to freeze the parameter.
>
>   3. Freezing $\mathbf{D}$ usually does not have a big impact, on either the training stability or the performance. This parameter introduces a skip connection and causes fewer issues (while also being less important) than other parameters.
>
>   We hope this summary is useful in answering the reviewer's question. We will include this discussion in our camera-ready submission if the paper is accepted.
>
> * ### Additional Comparison with Mamba2 (Q5)
>
>   We thank the reviewer for raising a discussion of Mamba2 as we believe this is an interesting model to compare with. To better compare B2S6 across different models, we will include the following table in our camera-ready submission if the paper is accepted.
>
>   | | S4D | S5 | Mamba | Mamba2 | B2S6 |
>   | --- | :---: | :---: | :---: | :---: | :---: |
>   | $\Delta_t$ Dependency | input-independent | input-independent | input-dependent  | input-dependent | input-dependent |
>   | $\mathbf{B}$ / $\mathbf{C}$ Dependency | input-independent | input-independent | input-dependent | input-dependent | input-dependent + input-independent |
>   | Number of Heads | $\geq 1$ | $\geq 1$ | $= 1$ | $\geq 1$ | $\geq 1$ |
>   | Parameters | complex | complex | real | real | complex
>   | Parameterization of $\mathbf{A}$ | diagonal | diagonal | diagonal | scalar | diagonal|
>
>
>   As we indicated in the discussion after Theorem 4 and 5, the multi-head design will enhance the expressiveness of the model and equip it with a gentler inductive bias. Given that, Mamba2 is expected to be more expressive than Mamba1. We have rerun the experiments in Figure 2 and 3 with Mamba2. While the results are hard to present without being able to upload figures, we confirm that Mamba2 performs quite as good as B2S6 in the experiment in Figure 2. This is not surprising because in this experiment, we remove the bias term from B2S6 to avoid "cheating," so Mamba2 is very close to B2S6. Mamba2's performance is in between that of Mamba1 and B2S6 in the experiment in Figure 3, which shows the advantage of using the bias term $\mathbf{B}_{\text{bias}}$ in B2S6.
>
>   Mamba2 still suffers from the stability issue, so when trained on LRA, one needs to reduce the learning rate, just as B2S6. Moreover, without complex parameterization, Mamba2 falls short significantly on the LRA benchmark. The following results were obtained during the rebuttal period.
>
>   | | ListOps | Text | Retrieval | Image | Pathfinder | Path-X | Avg. |
>   | --- | :---: | :---: | :---: | :---: | :---: | :---: | :---: |
>   | Mamba |38.02| 82.98| 72.14| 69.82| 69.26| 67.32| 66.59|
>   | Mamba2 |41.45| 86.09| 79.23| 71.96 | 75.45| 69.07 | 70.54 |
>   | B2S6 |63.85| 88.32| 91.44| 88.81| 95.93| 97.90| 87.71|
>
>   Moreover, we also pretrained the Mamba2 model following the design in Appendix G. The table below shows the perplexity results in addition to Figure 7. As shown in the table, B2S6 converges slightly slower initially than both Mamba and Mamba2, and caught up in the end. In this small example, B2S6 shows a matching performance.
>
>   |Steps | Mamba | Mamba2 | B2S6 |
>   | --- | :---: | :---: | :---: |
>   |2620 | 61.5441 | 59.6233 | 62.6133 |
>   |5241 | 42.7838 | 39.8305 | 44.3846 |
>   |7862 | 34.1740 | 31.9994 | 35.0440 |
>   |10483 | 29.7074 | 27.0987 | 30.3787 |
>   |13104 | 27.0171 | 25.2930 | 27.3891 |
>   |15725 | 25.2265 | 24.0179 | 25.2668 |
>   |18346 | 23.9789 | 23.1224 | 23.9370 |
>   |20967 | 23.0044 | 22.3695 | 22.8394 |
>   |23588 | 22.2368 | 21.7509 | 22.1322 |
>   |26209 | 21.6068 | 21.2480 | 21.5206 |
>   |28830 | 21.0785 | 20.8218 | 20.9773 |
>
>   We will incorporate these new results and discussion into our paper. We hope they enhance the comprehensiveness of our discussion.
>
> * ### Training a Large-Scale Model (Q2 + W1)
>
>   This is a very good point. While we spent most of the rebuttal period running additional experiments with Mamba2, we will start training some larger models on a large-scale dataset `SlimPajama-627B` and test them using `lm-eval-harness`, as Reviewer rPYn suggests, and be sure to incorporate these results into the camera-ready submission if the paper is accepted.
>
> * ### Minor Typo (W4)
>
>   Thank you also for pointing out the inconsistency on Line 157 and 159. We will change both into "$k$-th."
>
> Thank you again for your careful review and constructive comments. We are committed to improving this manuscript, and please let us know if you have any other questions!

---

> > ### Comment · Reviewer_y9vW · 2025-08-03
> >
> > Thanks for your detailed response. In pretrained models, Mamba2 appears to achieve lower perplexity than B2S6. Are there deeper insights on this? Since the Long Range Arena (LRA) benchmark doesn’t directly reflect a model’s language modeling capability.

---

> > > ### Author Response · Authors · 2025-08-03
> > > **Thank you for reviewing our rebuttal and following up!**
> > >
> > > Thank you for your careful review of our rebuttal! We believe that the faster convergence of Mamba2 is an interesting observation, and we outline our intuition below.
> > >
> > > First, we note that the largest perplexity gain for Mamba2 appears in the early training steps; by the end of the epoch, all three models reach comparable perplexities. We attribute this early-stage advantage primarily to the parameterization of $\mathbf{A}$. In Mamba2, $\mathbf{A}$ is parameterized as a scalar shared across all diagonal entries, whereas Mamba1/B2S6 learns each diagonal entry independently. This difference introduces a few potential effects:
> > >
> > > - **Simpler dynamics:** The scalar form enforces uniform evolution across all features, effectively reducing the number of degrees of freedom. This simplification can help the model capture coarse temporal patterns more quickly before learning more specialized dynamics.
> > >
> > > - **Better conditioning:** With a single learnable parameter, the scalar parameterization avoids issues arising from poorly conditioned dynamics across channels. This can lead to faster and more consistent updates in the early stages of training.
> > >
> > > These benefits appear to be mostly an early-training effect. As training progresses, the diagonal parameterization’s greater flexibility allows it to catch up in expressiveness, especially in tasks benefiting from channel-specific temporal modeling. This aligns with our observation that the final perplexity metric of scalar and diagonal forms can be similar.
> > >
> > > Understanding these points in-depth may require an independent research, but for a quick verification, we modified B2S6 to also use a scalar parameterization of $\mathbf{A}$ (as in Mamba2) to test our hypothesis empirically. While we did not run the experiment to the end of the epoch, the early-stage perplexities are:
> > >
> > > | Steps | B2S6 (with scalar $\mathbf{A}$) |
> > > |:-----:|:------------------------------:|
> > > | 2620  | 60.0285                         |
> > > | 5241  | 40.1730                         |
> > > | 7862  | 32.2596                         |
> > >
> > > This result supports the intuition that scalar parameterization accelerates initial convergence. We are happy to include this discussion in our paper as a possible future research direction.

---

> > > > ### Comment · Reviewer_y9vW · 2025-08-04
> > > >
> > > > Thank you for the additional clarifications. This is an exceptionally valuable study. I recommend accepting the paper.

---

> > > > > ### Author Response · Authors · 2025-08-04
> > > > >
> > > > > Thank you once again for your insightful feedback and your ongoing support in our paper!

---

### Official Review · Reviewer_8YXd · 2025-07-02

**Clarity:** 4
**Significance:** 3
**Originality:** 3
**Rating:** 5
**Confidence:** 4

**Summary:**

Driven from the concrete motivation -- why Mamba performs poorly on long range tasks -- the paper points out a simple solution: use of bias. The authors theoretically investigate critical limitations (supported by proofs) of S6 such as less expressivity with a single layer than S4D, strong inductive bias which can be particularly harmful for long sequences,  and instability. The paper empirically validates that simply adding group-wise (similar to Mamba2) input-independent bias, S6 becomes much stronger in long range modeling while not hurting the language modeling performance.

**Questions:**

Does the same effects arise in Mamba2 SSD kernel too? What's the main reason of choosing S6 to compare against instead of SSD?

**Ethical Concerns:**

["NO or VERY MINOR ethics concerns only"]

**Final Justification:**

Most of my concerns are addressed during the rebuttal, thus I maintain my score.

**Limitations:**

yes

**Paper Formatting Concerns:**

no concerns

**Quality:**

4

**Strengths And Weaknesses:**

The paper is very well written, and I really appreciated its strong motivation and simple solution. In high-level, the addition of bias harmonizes the strengths of both S4D and S6, making Mamba become much more powerful. The claims are well supported in both theoretical and empirical validations, making the paper more credible. A weakness (which is a nitpick) is lack of evaluation on language modeling benchmarks (and the training scale is too small: 6B tokens), and it would be interesting to see if this simple change can lead to gains in real-world language modeling capacities such as retrieval.

---

> ### Author Rebuttal · Authors · 2025-07-30
>
> We thank the reviewer for the careful review and insightful comments.
>
> We chose Mamba1 as a reference model mainly because it is the first selective state-space model proposed. Nonetheless, we are happy to include additional comparison of our B2S6 model with Mamba2. First, conceptually, as suggested in section 6 of the paper, the multi-head structure is implemented in Mamba2, so our main contributions are theoretical analysis of the multi-head structure, together with a new design of a bias term $\mathbf{B}_{\text{bias}}$. We will add the following table to our manuscript to further compare the design choices of different state-space models in a clear way:
>
> | | S4D | S5 | Mamba | Mamba2 | B2S6 |
> | --- | :---: | :---: | :---: | :---: | :---: |
> | $\Delta_t$ Dependency | input-independent | input-independent | input-dependent  | input-dependent | input-dependent |
> | $\mathbf{B}$ / $\mathbf{C}$ Dependency | input-independent | input-independent | input-dependent | input-dependent | input-dependent + input-independent |
> | Number of Heads | $\geq 1$ | $\geq 1$ | $= 1$ | $\geq 1$ | $\geq 1$ |
> | Parameters | complex | complex | real | real | complex
> | Parameterization of $\mathbf{A}$ | diagonal | diagonal | diagonal | scalar | diagonal|
>
> On the experimental side, we have trained Mamba2 models on the LRA benchmark during the rebuttal period, and the following table summarizes the results:
>
> | | ListOps | Text | Retrieval | Image | Pathfinder | Path-X | Avg. |
> | --- | :---: | :---: | :---: | :---: | :---: | :---: | :---: |
> | Mamba |38.02| 82.98| 72.14| 69.82| 69.26| 67.32| 66.59|
> | Mamba2 |41.45| 86.09| 79.23| 71.96 | 75.45| 69.07 | 70.54 |
> | B2S6 |63.85| 88.32| 91.44| 88.81| 95.93| 97.90| 87.71|
>
> Moreover, we also pretrained the Mamba2 model following the design in Appendix G. The table below shows the perplexity results in addition to Figure 7. As shown in the table, B2S6 converges slightly slower initially than both Mamba and Mamba2, and caught up in the end. In this small example, B2S6 shows a matching performance.
>
> |Steps | Mamba | Mamba2 | B2S6 |
> | --- | :---: | :---: | :---: |
> |2620 | 61.5441 | 59.6233 | 62.6133 |
> |5241 | 42.7838 | 39.8305 | 44.3846 |
> |7862 | 34.1740 | 31.9994 | 35.0440 |
> |10483 | 29.7074 | 27.0987 | 30.3787 |
> |13104 | 27.0171 | 25.2930 | 27.3891 |
> |15725 | 25.2265 | 24.0179 | 25.2668 |
> |18346 | 23.9789 | 23.1224 | 23.9370 |
> |20967 | 23.0044 | 22.3695 | 22.8394 |
> |23588 | 22.2368 | 21.7509 | 22.1322 |
> |26209 | 21.6068 | 21.2480 | 21.5206 |
> |28830 | 21.0785 | 20.8218 | 20.9773 |
>
> We hope these experiments enhance the comprehensiveness of the manuscript. We also agree with the reviewer that the paper would benefit more from larger-scale experiments on language datasets. To this end, we will start training some larger B2S6-based LLMs on a large-scale dataset `SlimPajama-627B` and be sure to incorporate these results in the camera-ready submission if the paper is accepted.
>
> Thank you again for your careful review and constructive comments. We are committed to improving this manuscript, and please let us know if you have any other questions!

---

> > ### Comment · Reviewer_8YXd · 2025-08-04
> > **Official Comment by Reviewer 8YXd**
> >
> > Thank you for your clarifications. Most of my concerns are addressed, thus I will keep my score.

---

> > > ### Author Response · Authors · 2025-08-04
> > >
> > > Thank you once again for your insightful feedback and your ongoing support in our paper!

---

### Official Review · Reviewer_rPYn · 2025-07-03

**Clarity:** 3
**Significance:** 3
**Originality:** 3
**Rating:** 4
**Confidence:** 4

**Summary:**

The paper presents a study of current state space models (SSMs) like S4D and Mamba. As a result a number of weka points are discovered: training stability, poor long-range performance, inductive bias and expressiveness. To compensate for these shortcomings, a new architecture is proposed named as B2S6. The proposed architecture combines block-wise selective dynamics (as in Mamba1) with a channel-specific bias. Most evaluations are done on Long-Range Arena.

**Questions:**

- How does it perform in the hybrid settings (combined with Attention)? Do we need a new head design, or attention can compensate for it?
- Propose to create a table with a summary of differences between S4D, Mamba, B2S6 and Mamba2.
- For Figure 2, can the failure of Mamba be explained by not optimal learning rate, or should we increase number of heads even more?
- What is the impact on speed of using B2S6 vs Mamba1/2?

**Ethical Concerns:**

["NO or VERY MINOR ethics concerns only"]

**Limitations:**

Limitations are mentioned in the Supplementary Materials.

**Quality:**

3

**Strengths And Weaknesses:**

- The paper contains a comprehensive analysis of the current state of Mamba1.
- Key limitations of Mamba's ability to perform well on long-range dependencies are summarized as a lack of expressiveness, inductive bias, and training stability.
- The background and recap of S4D and S6 are well covered in Section 2.
- The analysis provided in Section 3 is interesting and highlights the insights mentioned above. Including Table 1 and Figure 2 is great.
- Section 4 is well organized.
- Section 5 analyzes the training stability of Mamba. From personal experience, we have not encountered issues when training models with Mamba1. How robust are the conclusions in this section to applying weight decay or using pseudo second-order optimization frameworks like Adam?
- B2S6, introduced in Section 6, appears novel and well-grounded. However, more discussion is needed regarding comparisons to Mamba2. Lines 253–255 suggest that the proposed architecture is very close to Mamba2, though the authors claim to provide additional insights. Is this accurate?


Weaknesses:
1) The experimental analysis is weak—it takes less than one page in the main paper. It is important to assess whether B2S6 scales to LLMs and whether observations from the small-scale model will generalize to larger models. Training instabilities might emerge.
2) The intuition behind incorporating per-channel separation in B2S6 is related to the Mamba2 extension proposed in [20]. However, the current evaluation lacks comparison to [20].
3) Overall, the paper lacks a deeper analysis of the proposed model and B2S6. For example, Mamba2 should be included in Figure 2 and Table 1.
Ideally, B2S6 should be evaluated in an LLM setting (at least a 1–2B parameter model trained on 200B tokens) and compared to other works mentioned in the paper, such as Mamba1. Current evaluations are insufficient. I recommend the authors use lm-eval-harness to obtain more comprehensive evaluation benchmarks.

---

> ### Author Rebuttal · Authors · 2025-07-30
>
> We thank the reviewer for the careful review and insightful comments. Below we address the questions and comments raised in this review.
>
> * ### Multi-head Design in a Hybrid Model (Q1)
>
>   This is a very interesting question. In a hybrid model, such as Jamba, Mamba layers and attention layers induce different biases and are used to capture different dependencies in the sequence of tokens (e.g., attention is good at handling short-term, high-salience interactions while Mambas are good for long-range dependencies). That said, from a universal approximation perspective, the attention layers alone are capable of learning most smooth ground truth functions; in practice, however, the multi-head design in attention layers cannot totally compensate for the lack of heads in a Mamba layer in order to better learn all aspects of the sequence.
>
>   Another interesting thing to point out is that, even though the word "multi-head" is used for both attention and Mamba, the precise mechanisms are slightly different. For a multi-head attention, the input into each "head" is the entire sequence; hence, increasing the number of heads mainly introduces a trade-off between the variance and the computational time. For Mamba, however, the input into each "head" is only a sub-block of the sequence, consisting of a subset of channels, and this induces a trade-off between the variance (in terms of the number of different time-variant systems we have) and bias (in terms of how much we can achieve with each system). Hence, we advocate an independent multi-head design for Mamba layers and attention layers in a hybrid model.
>
> * ### Conceptual Comparison with Mamba2 (Q2 + S7)
>
>   Thank you for your comment. We agree that a table will improve the clarity of our method. We plan to include the following table to summarize the differences between each state-space model.
>
>   | | S4D | S5 | Mamba | Mamba2 | B2S6 |
>   | --- | :---: | :---: | :---: | :---: | :---: |
>   | $\Delta_t$ Dependency | input-independent | input-independent | input-dependent  | input-dependent | input-dependent |
>   | $\mathbf{B}$ / $\mathbf{C}$ Dependency | input-independent | input-independent | input-dependent | input-dependent | input-dependent + input-independent |
>   | Number of Heads | $\geq 1$ | $\geq 1$ | $= 1$ | $\geq 1$ | $\geq 1$ |
>   | Parameters | complex | complex | real | real | complex
>   | Parameterization of $\mathbf{A}$ | diagonal | diagonal | diagonal | scalar | diagonal|
>
>   Also, we confirm that our discussion on Line 253–255 suggests that the multi-head design is not new to SSMs, but in this paper we provide further theoretical analysis to motivate it.
>
> * ### Additional Experimental Comparison with Mamba2 (W2 + W3)
>
>   We are happy to run some experiments to incorporate Mamba2 in our comparison. The following table compares Mamba2 with Mamba1 and B2S6 on the LRA benchmark:
>
>   | | ListOps | Text | Retrieval | Image | Pathfinder | Path-X | Avg. |
>   | --- | :---: | :---: | :---: | :---: | :---: | :---: | :---: |
>   | Mamba |38.02| 82.98| 72.14| 69.82| 69.26| 67.32| 66.59|
>   | Mamba2 |41.45| 86.09| 79.23| 71.96 | 75.45| 69.07 | 70.54 |
>   | B2S6 |63.85| 88.32| 91.44| 88.81| 95.93| 97.90| 87.71|
>
>   Moreover, we also pretrained the Mamba2 model following the design in Appendix G. The table below shows the perplexity results in addition to Figure 7. As shown in the table, B2S6 converges slightly slower initially than both Mamba and Mamba2, and catches up in the end. In this small example, B2S6 shows a matching performance.
>
>   |Steps | Mamba | Mamba2 | B2S6 |
>   | --- | :---: | :---: | :---: |
>   |2620 | 61.5441 | 59.6233 | 62.6133 |
>   |5241 | 42.7838 | 39.8305 | 44.3846 |
>   |7862 | 34.1740 | 31.9994 | 35.0440 |
>   |10483 | 29.7074 | 27.0987 | 30.3787 |
>   |13104 | 27.0171 | 25.2930 | 27.3891 |
>   |15725 | 25.2265 | 24.0179 | 25.2668 |
>   |18346 | 23.9789 | 23.1224 | 23.9370 |
>   |20967 | 23.0044 | 22.3695 | 22.8394 |
>   |23588 | 22.2368 | 21.7509 | 22.1322 |
>   |26209 | 21.6068 | 21.2480 | 21.5206 |
>   |28830 | 21.0785 | 20.8218 | 20.9773 |
>
>   We also reran the experiment in Figure 2 with Mamba2. While the results are hard to deliver in a table, we confirm that Mamba2, unlike Mamba1, does benefit from an increasing hidden dimension $d$. We provide more explanation about this below, and if our paper is accepted, we will include all these results in the camera-ready submission.
>
> * ### Training a Large-scale Model (W1)
>
>   This is a very good point. While we spent most of the rebuttal period running additional experiments with Mamba2, we will start training some larger models on a large-scale dataset `SlimPajama-627B` and evaluate them using `lm-eval-harness` and be sure to incorporate these results in the camera-ready submission if the paper is accepted.
>
> * ### Explaining Figure 2 (Q3)
>
>   This is an interesting point, and we have done some further hyperparameter tuning over a grid of 10 learning rates, ranging from $\texttt{1e-4}$ to $\texttt{1e-1}$. We observed that none of these learning rates are able to save S6's performance on this task. Hence, we conclude that this is mainly an issue with a lack of the number of heads. Note that the only difference between models in the middle panel and right panel is that the B2S6 model has more heads than Mamba, which is a single-head model. (On Line 271-272, we clarify this by noting that "we revisit the experiment from section 3, this time using only the block structure (without the bias term).")
>
>   Another interesting point to make here is that the S4D model performs better than the B2S6 model on this task, and this is true even if we set the number of heads in B2S6 to be equivalent to the hidden dimension $d$. The reason is that the LTI systems in an S4D model are intrinsically more capable of learning a single Fourier mode (see [96] in the bibliography). Hence, the primary lesson from Figure 2 is not that S4D is better than S6 because it has more "heads," but that increasing the number of heads from 1 will enhance Mamba's expressiveness. We will make this clear in our paper.
>
> * ### Efficiency Comparision (Q4)
>
>   We report the training throughputs (tokens/sec) of Mamba and B2S6 models. We remark that the code we used for training the Mamba and B2S6 models is purely PyTorch-based and does not involve any CUDA optimization. We mentioned this in the limitation section, and once the CUDA optimization is incorporated into the B2S6 implementation, its efficiency will be improved.
>
>   | Model Size | Mamba | B2S6 |
>   | :--- | :---: | :---: |
>   | 250M | 14,667 | 14,218 |
>   | 370M | 9,914 | 9,676 |
>   | 500M | 7,218 | 7,010 |
>
> * ### Applicability of the Stability Analysis (S6)
>
>   This is another interesting thing to comment on. In training a Mamba-based LLM, we indeed have not observed instability induced by $\Delta$. However, when training Mamba models on LRA benchmark tasks, without carefully reducing the learning rate of $\Delta$, the training is very volatile. One reason is that the models we train on LRA are much smaller than LLMs (and hence, the loss landscapes are less smooth than those of a larger model), and the sequences in LRA tasks can also be very long, e.g., in Path-X, a sequence has a length of 16,384. This is very large with respect to the size of the model we trained.
>
>   You are absolutely right to think of Adam and weight decay as ways to speed up the convergence and stabilize the training. Our analysis in Theorem 3, on the other hand, is independent of the optimizer or the learning rate scheduler. Since we aim to analyze the stability without having a specific loss function, dataset, optimizer, or scheduler in mind, we go for the most general strategy and investigate the gradient norm. In general, a larger gradient norm leads to less stable training, and this holds for any optimizer and scheduler. In that sense, the Adam optimizer, weight decay, and potentially gradient norm clipping and learning rate reduction can all be viewed as efforts to counterbalance the stability issue raised in Theorem 3.
>
> Thank you again for your careful review and constructive comments. We are committed to improving this manuscript, and please let us know if you have any other questions!

---

### Comment · Area_Chair_5mEa · 2025-08-04

Dear reviewers,

Thank you for your valuable time and your expertise in reviewing. Engaging with authors is really important, and allows both authors and reviewers to gain deeper understanding of cutting edge topics. This is a unique opportunity of interaction in our community.

The author rebuttal phase is about to close, and we kindly request your prompt attention to ensure a thorough discussion.

The **discussion period ends in less than 3 days** (on Aug. 6, 11:59pm AOE ). To maintain the review timeline, we ask that you:

- Review the rebuttals,

- Engage in any ongoing discussion with fellow reviewers/authors (if applicable),

- Finalize your assessment.

If you have already completed this step, we sincerely appreciate your efforts.

Thank you for your collaboration!

Best regards,

AC

---

### Note · Authors · 2025-08-13

Dear AC and reviewers,

We sincerely thank all reviewers for their constructive feedback and thoughtful discussions during the rebuttal period. In addition to addressing each reviewer’s specific questions, we take this opportunity to summarize key points raised by the reviewers and the corresponding improvements we have made or planned:

* ### Comparison to Mamba2
Several reviewers asked how our B2S6 model compares to Mamba2. To address this, we will add a clear comparison table in the manuscript outlining the distinctions between S4D, S5, S6 (Mamba), Mamba2, and B2S6. This table is already included in our rebuttal for reference. We also conducted additional experiments with Mamba2, including the Long Range Arena benchmark and language modeling on SlimPajama-6B. These new results, which can be found in our rebuttals below, are consistent with our central claims and strengthen our empirical comparisons.

* ### Additional Experiments with B2S6
We performed further efficiency measurements for B2S6 and, following reviewer suggestions, have planned a larger-scale training run to explore its performance at scale. These results will be included in the camera-ready version of the paper if it is accepted.

* ### Rephrasing Theorems and Discussions
Some reviewers noted a few presentation issues in our theorem discussions. We have outlined specific revisions, detailed in our response to reviewer PVYx, that improve clarity, rigor, and fairness in the theoretical discussion.

We thank all reviewers for their great comments and believe that these updates will further strengthen the manuscript. Thank you again for your consideration and for supporting the refinement of this work!

---

### Decision · Program_Chairs · 2025-09-17

**Decision:**

Accept (poster)

**Comment:**

The paper at hand proposes a few modifications to the Mamba architecture targeted to improving expressivity and optimization properties of the block. Going beyond the original Mamba design, keeping efficiency while improving long-range capabilities is a highly relevant, timely topic. I particularly enjoyed the careful notation and precision of the paper, as well as the thorough ablations.


All reviewers agree that the paper deserves publication, and the authors provided additional experiments supporting their claims during the rebuttal period. For this reason, I recommend accepting this work.